# Beyond Token Probes: Hallucination Detection via Activation Tensors with ACT-ViT

**Guy Bar-Shalom**[*]
Technion
guybs99@gmail.com

**Fabrizio Frasca**[*]
Technion
fabriziof@campus.technion.ac.il

**Yaniv Galron**
Technion

**Yftah Ziser**
University of Groningen, Nvidia Research

**Haggai Maron**
Technion, Nvidia Research

## Abstract

Detecting hallucinations in Large Language Model-generated text is crucial for their safe deployment. While probing classifiers show promise, they operate on isolated layer–token pairs and are LLM-specific, limiting their effectiveness and hindering cross-LLM applications. In this paper, we introduce a novel approach to address these shortcomings. We build on the natural sequential structure of activation data in both axes (layers × tokens) and advocate treating full activation tensors akin to images. We design ACT-ViT, a Vision Transformer-inspired model that can be effectively and efficiently applied to activation tensors and supports training on data from multiple LLMs simultaneously. Through comprehensive experiments encompassing diverse LLMs and datasets, we demonstrate that ACT-ViT consistently outperforms traditional probing techniques while remaining extremely efficient for deployment. In particular, we show that our architecture benefits substantially from multi-LLM training, achieves strong zero-shot performance on unseen datasets, and can be transferred effectively to new LLMs through fine-tuning. Full code is available at `https://github.com/BarSGuy/ACT-ViT`.

## 1 Introduction

Despite their remarkable performance across a wide range of tasks, the inner workings of Large Language Models (LLMs) remain poorly understood. This lack of transparency is particularly problematic in high-stakes scenarios, where LLMs are prone to "hallucinations" – cases in which the model generates false or fabricated content [62, 38, 22, 26, 54]. Accurate Hallucination Detection (HD) is critical for the safe and reliable deployment of LLMs [51, 35, 55]. While existing methods for detecting such hallucinations often rely on auxiliary LLMs or repeated prompting [34, 48], these approaches tend to be slow and computationally intensive (up to several seconds per instance [6]). A prominent class of methods suggest to overcome this additional computational cost via probing classifiers: simple models trained on internal LLM representations to predict certain attributes [7]. These techniques have been central to interpretability research [20, 31, 58, 15, 1, 13, 8], with linear probes gaining popularity for their transparency and simplicity [4, 25, 37, 44, 48].

Unfortunately, traditional probing classifiers exhibit two fundamental limitations. First, they operate on isolated layer–token pairs, neglecting information from other activations within the model. This is critical: while predictive hallucination signals might indeed appear at specific "exact tokens", their positions vary across responses, making the position-finding a difficult problem on its own—practically requiring (multiple) LLM queries [48]. In addition to this, these signals may also peak at different

---

[*]Equal contribution

39th Conference on Neural Information Processing Systems (NeurIPS 2025).

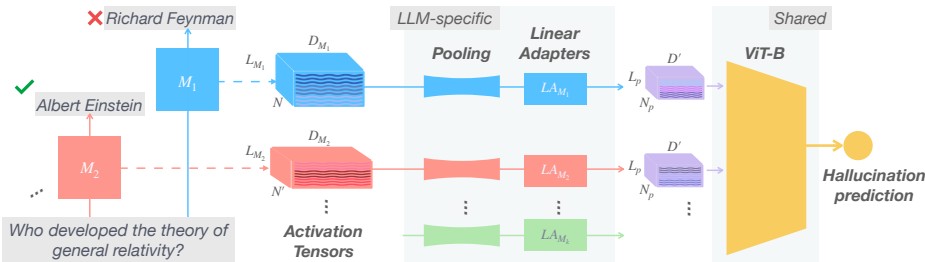

Figure 1: `ACT-ViT` **overview:** we extract Activation Tensors from (multiple) LLMs, apply Pooling, and project them to a shared space via per-LLM Linear Adapters. A shared ViT Backbone is then applied. `ACT-ViT` benefits from training on data from multiple LLMs and can be easily fine-tuned to unseen ones.

layers on a per-dataset basis [5], underscoring the need for a more holistic processing of the model's internal state. A second limitation is that probing classifiers are model-specific, thus unable to generalize across LLMs or to benefit from hallucination-related insights found in other LLMs. This hinders sharing datasets across LLMs, transfer learning, and multi-model training. Crucially, each new LLM or text generation task demands annotating a new dataset, a tricky and sometimes impractical procedure. Motivated by the above, we ask: *(i) Can we design an architecture that processes the full internal state of an LLM and benefit from it?*; and *(ii) Can it learn across multiple LLMs and benefit from it?* In this paper, we positively answer these questions and propose a novel approach, dubbed `ACT-ViT`, addressing the aforementioned limitations in an effective and efficient way.

We start by providing further evidence on the limitations of single-position, static probing classifiers. On top of previous observations suggesting that the best probe-position varies across tokens and layers on a per-sample and dataset basis [48, 5], we show that the location of predictive signals for hallucinations also widely varies across LLMs, even for the same dataset. This further calls for the development of a more sophisticated approach that can adaptively attend to the most predictive "locations" within models' hidden states, especially if aiming to operate beyond the single-LLM regime. To this end, we propose leveraging, as our basic data type, what we term the *Activation Tensor* (AT) – a comprehensive representation of the LLM's internal state with respect to a given input. Formally, for an LLM $M$ and a $N$-long response to some input query, the AT is denoted $\mathbf{A} \in \mathbb{R}^{L_M \times N \times D_M}$, with $L_M$ the number of layers, $D_M$ $M$'s hidden dimension. ATs are informative but challenging to process: they are large, high-dimensional, and LLM-dependent. We process them efficiently and effectively by exploiting their structural similarity to images. An RGB image forms a $H \times W \times 3$ tensor with spatial dimensions (a grid of pixels) and feature channels. Similarly, Activation Tensors have "spatial" dimensions described by layer stacks ($L_M$) and token sequences ($N$), and a feature dimension corresponding to the dimensions of the LLM's hidden states. This analogy enables applying proven computer vision techniques like spatial pooling and Vision Transformers [14].

To unify ATs across different LLMs into a consistent shape, we apply pooling over the spatial dimensions, followed by an LLM-specific Linear Adapter (LA) on the feature dimension. This transforms all tensors into a shared shape of $L_p \times N_p \times D'$. The resulting representation is then fed into a *shared* ViT-based backbone. See Figure 1 for a complete depiction of our framework. These components are jointly trained for HD on multiple annotated AT datasets originating from different LLMs. Importantly, our use of linear adapters is inspired by recent work [24, 46] suggesting that LLMs converge towards a shared statistical model of reality, with the notion of universal truthfulness [43, 60] positing they learn a generalizable, linearly exploitable representation of factual truth. A key aspect of our architecture is its ability to be applied to new, unseen LLMs. In these cases, it suffices to only train the lightweight, LLM-specific LA, while the backbone – pretrained for HD in a unified representation space across LLMs – is kept frozen.

**Results.** In our extensive experiments across 15 LLM–dataset combinations, `ACT-ViT` consistently outperforms traditional probing methods in both single- and multi-LLM training settings. Remarkably, it exhibits strong zero-shot generalization to *new datasets* from LLMs seen during training and requires as little as $\approx 5\%$ of the training data to improve performance further and – in many cases – surpass traditional probes trained on the whole dataset. Furthermore, by training *only* the lightweight LA for a *new unseen LLM*, our model achieves superior performance than standard probes. `ACT-ViT`

trains on a single GPU in less than three hours on all 15 LLM-dataset combinations together, and detects hallucination in less than $\approx 10^{-5}$ seconds per instance at inference time.

**Contributions.** (**1**) We propose Activation Tensors (ATs) as a holistic, structured representation of LLM internals. (**2**) Inspired by the similarity of ATs to images we introduce a novel architecture that can effectively train on ATs from multiple LLMs and datasets. We show it outperforms traditional probes, even when trained only on a single LLM. (**3**) We demonstrate strong transfer capabilities: zero-shot generalization to new datasets on LLMs that were seen during training, and fast adaptation to out-of-domain LLMs with only a newly trained linear adapter, keeping the remaining parameters frozen. (**4**) We show our method is highly efficient during both training and inference.

## 2 Related Work

**Probing classifiers.** In NLP, probing classifiers [7] are used to analyze model representations and reveal how features like syntax and semantics are encoded [19, 12, 30]. Originating in interpretability research, they help identify which model components capture specific linguistic properties. E.g., lower layers capture local syntax in machine translation, while higher layers encode global structure [57]. Probes typically operate at the token and layer levels, with sentence-level tasks often using the final token's activation [1, 29, 61]. While both linear probes and MLPs are common [4, 25, 37, 44], linear probes are preferred for interpretability, as MLPs are often too expressive to distinguish whether the structure lies in the embeddings or the probe itself [21]. Although early interpretability studies favored simpler probes, more recent work uses probing as a predictive or control tool and prioritize prediction efficacy. Probes have recently estimated LLMs' downstream performance [70, 2, 16] and behavioral traits, e.g. sycophancy [49], or have guided models to desirable behaviors, e.g. reduced bias or improved factuality [65, 52, 41]. Still, many approaches use minimal setups, such as a single token from a single layer fed into a linear probe. Our work moves beyond these constraints by designing high-performing probes that better and more comprehensively leverage internal representations.

**Error Detection in LLMs.** LLMs produce diverse and complex mistakes and identifying these has become increasingly important. Hallucinations—a well-studied subset—are typically defined as outputs unfaithful to the input or external facts. Yet their definitions vary, extending to biases, reasoning flaws, and other failures [38, 22, 26, 54]. We adopt a broader view of error detection to account for this ambiguity, treating hallucinations as one specific case. While we occasionally use the terms interchangeably, we recognize that hallucinations are only one facet of the broader error landscape. Most prior work approaches error detection using uncertainty estimates [29, 64, 33, 42] or probing internal representations—often focusing on the final answer token [29, 61, 69, 71, 68, 11, 59, 36, 43, 10, 53]. Others probe the final prompt token to anticipate errors before generation [60, 61, 59, 17, 56]. While effective, these methods under-exploit the richness of model activations. We address this by developing stronger methods that make fuller use of internal representations.

## 3 Notation, Problem Formulation and Setup

Given LLM $M$, input query $\vec{s}$ and the LLM-generated response $\vec{g}$, the goal is to predict whether $\vec{g}$ is correct or not. We assume a white-box setting, i.e. full access to the internal states of $M$, but disallow using external resources (such as search engines or auxiliary LLMs). We restrict the setting to single-pass prompting with no rounds of interactions due to their high cost.

**Activation Tensors.** We define the *activation tensors* as the hidden representations within the LLM across all layers and all tokens in the output text.[2] Formally, the Activation Tensor (AT) of an LLM $M$ is a third-order tensor $\mathbf{A} \in \mathbb{R}^{L_M \times N \times D_M}$, where $L_M$ is the number of layers in the model, $N$ is the output sequence length, and $D_M$ is the feature (hidden state) dimension; see Figure 1. Throughout this paper, we focus on the hidden states at each layer after the application of the feed-forward network and the residual connection, refer to Appendix A for details.

**Dataset construction.** Following the setup of [48], we consider datasets $\mathcal{D} = \{(\vec{s}_i, \vec{g}_i)\}_{i=1}^{m}$, consisting of $m$ question-answer pairs, such as: ("*Who developed the theory of general relativity?*",

---

[2]This definition can be readily extended to include the input text as well.

"*Albert Einstein*"). For each query $\vec{s}_i$, the model produces a response $\vec{\tilde{g}}$, and we extract the associated activation tensor $\mathbf{A}_i$, yielding a collection $\{\mathbf{A}_i\}_{i=1}^m$. We calculate binary *hallucination labels*, by comparing each response $\vec{\tilde{g}}_i$ with answer $\vec{g}_i$ to determine its correctness. We assign label $y_i = 1$ for a correct response, $y_i = 0$ otherwise. Our final *activation dataset* is $\mathcal{A} = \{(\mathbf{A}_i, y_i, M_i)\}_{i=1}^m$, where $M_i$ is the LLM producing AT $\mathbf{A}_i$ associated with hallucination label $y_i$. We importantly note that multiple distinct $M_i$'s from a known collection $\mathcal{M}$ are allowed within the same dataset.

**Problem Statement.**   Our objective is to train a parametric model mapping ATs to their corresponding hallucination labels. We additionally consider the following more general and challenging scenarios: (i) *Off-domain data generalization:* ATs are drawn from a known model $M \in \mathcal{M}$, but correspond to tasks or domains not seen during training (e.g., training on $M$'s activation tensors from sentiment analysis, but testing on trivia question answering). (ii) *Off-domain model generalization:* ATs are drawn from a previously unseen LLM $M \notin \mathcal{M}$.

# 4   Towards Better, Non-Static Probing

In the context of a single, target LLM, Orgad et al. [48] uncovered the importance of probing on "exact-token" position which may vary significantly on a per-sample basis – and generally requires external algorithms, such as multiple expensive queries to an LLM, in order to be identified. Complementary to this, the results in [5] suggest a degree of variability in the position of the best probing location also across layers. We consolidate these observations and further extend them by examining cross-LLM variability for fixed target datasets, further underscoring the need to go beyond static probing.

Inspired by the analyses by Orgad et al. [48] on Mistral-7B-Instruct-v0.2 [27] (Mis-7B), we train distinct logistic regression classifiers for each layer–token combination and construct ROC-AUC heatmaps to visualize the presence and localization of hallucination signals. We conduct this analyses across three distinct LLMs and five different datasets. Since sentence lengths vary, we restrict tokens to $\mathcal{N} := \{0, 1, 2, -3, -2, -1\}$, namely the first and last three in each sentence. Heatmaps are presented in Figure 2 for Mis-7B,

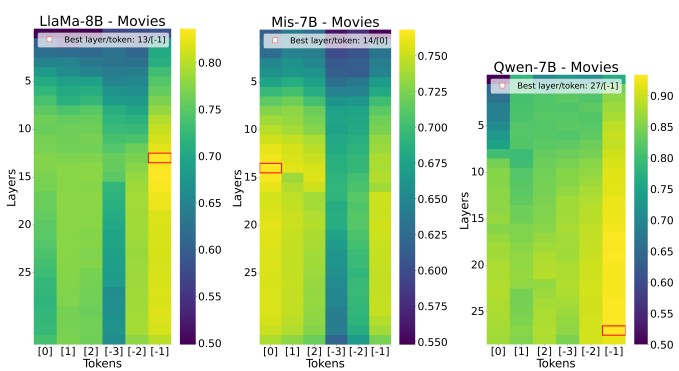

Figure 2: Test AUC heatmaps across layer–token combinations; best layer-token combinations are boxed.

Llama-3-8B-Instruct [63] (LlaMa-8B) and Qwen2.5-7B-Instruct [66] (Qwen-7B), over the Movies dataset [48] (we refer to Appendix A.5 for their full set). Figure 2 shows predictive signals spread over multiple layer-token combinations, but it clearly indicates that the most informative positions vary significantly across the three different LLMs, with distinct patterns characterizing each heatmap. In particular, the most predictive signal in terms of layers appears around the middle of the model for LlaMa-8B and Mis-7B, whereas for Qwen-7B it is concentrated in the final layers. As for token positions, the signal tends to be strongest at the end of the sentence for both LlaMa-8B and Qwen-7B, while for Mis-7B it is more prominent towards the beginning. Concretely, note how the best probing position for Mis-7B, i.e., (14, 0), delivers suboptimal results on both LlaMa-8B and Qwen-7B, where the gap with their respective best probing locations is $\approx 7\%$ AUC on both.

**Towards Better Probing Techniques.**   Traditional probing methods rely on fixed layer/token selections. We claim this limits their ability to capture the dynamic nature of error signals, whose position vary across layers and tokens. This rigidity also hampers generalization across LLMs and tasks. We believe that a more effective approach should adaptively attend to relevant activations while remaining efficient. To this end, we introduce – and present next – an architecture that processes the full AT $\mathbf{A}$, enabling a flexible and powerful HD across models and datasets.

### 4.1 Learning on Activation Tensors

Our starting observation is that Activation Tensors $\mathbf{A} \in \mathbb{R}^{L_M \times N \times D_M}$ are highly *structured* objects: they span two *sequential* dimensions and a *feature* dimension. The former two correspond to LLM layers ($L_M$) as well as generated tokens ($N$), forming a structural map of neural activity across text positions and processing depths. The latter corresponds, instead, to the dimension of the LLM's hidden states ($D_M$). We note that this structure is remarkably analogous to that of images: layers and tokens are in correspondence with their vertical and horizontal spatial dimensions, the hidden states with channels (e.g., RGB). Based on this analogy, we refer to the layer-token dimensions ($L_M, N$) as the *spatial activation dimensions*, and each layer-token location as an *activation pixel*. Accordingly, we propose to process ATs similarly to images, designing our method using techniques inspired by modern vision architectures and guided by the principles outlined below.

**Guiding principles.** We follow two key principles in designing architectures for Activation Tensors: (i) cross-LLM generalization and (ii) computational efficiency. Cross-LLM capability is especially crucial for hallucination detection, where annotated data is scarce, costly, or unavailable for new models. We aim to build a model that learns from multiple LLMs and generalizes effectively to unseen ones, without incurring high computational cost. Principle (ii) poses a major challenge: Activation Tensors are large. For example, a transformer with $L = 50$ layers, $N = 512$ tokens, and $D = 4096$ hidden dimensions produces a single AT of $\approx 0.2$GB in `float16`; a batch of 64 totals $\approx 12.8$GB. Addressing this scale is critical for real-world use. The next section introduces `ACT-ViT`, an architecture designed to handle ATs efficiently while meeting the principles above.

### 4.2 `ACT-ViT`

**Overview.** `ACT-ViT` consists of the following components (see Figure 1): (1) a *Pooling* (Pool) layer that reduces the "spatial" dimensions of the activation tensor (layers and tokens) to a fixed, predefined size ($L_p, N_p$); (2) a per-LLM *Linear Adapter* (LA) layer that compresses and "aligns" their feature dimensions; (3) a *ViT-Based Backbone* (ViT-B), which processes the resulting tensor using a Vision Transformer architecture, enhanced with shared positional encodings within each patch. In what follows, we expand on each of these components.

**Pool.** Similarly to image resizing, we propose to reduce the size of activation tensors before processing them with our architecture: as a preprocessing step, we apply a max-pooling operation across the spatial activation dimensions (see inset illustration). Specifically, we propose two possible pooling approaches. The first maintains full input resolution at the level of tokens and applies *1d* pooling on the layer dimension only. This approach

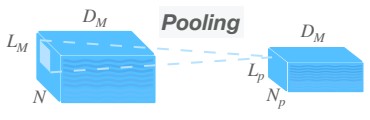

is preferred when expecting relatively concise LLM generations, or when their maximum generation length remains computationally manageable. Alternatively, a second approach adopts a more general *2d* pooling on both the two spatial activation dimensions, allowing to more efficiently deal with potentially longer LLM responses. As we will detail next, we will mainly adopt the first approach, and additionally experiment with the second in Section 5.

Note that, in our applications, not only the number of generated tokens $N$ naturally varies depending on the input prompt, also, the number of LLM layers $L$ may vary across models in $\mathcal{M}$. With guiding principle (i) in mind, it becomes important to ensure our architecture can robustly deal with both these varying parameters. Accordingly, we design the two above pooling operations in a way that, regardless of the input size, their outputs, $\mathbf{A}^p \in \mathbb{R}^{L_p \times N_p \times D_M}$, always match a predefined shape of ($L_p, N_p$) in their spatial dimensions (see Algorithm 1, in Appendix B.). Importantly, this property ensures the downstream ViT backbone is always fed with tensors of a consistent spatial shape.

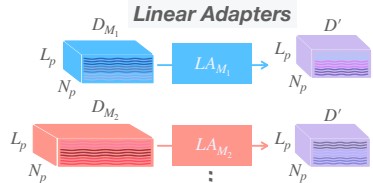

**LA.** Following pooling, we map the feature dimensions to a shared hidden size via a linear transformation. In particular, we propose to instantiate and employ a *dedicated* linear adapter (LA) module for each LLM $M \in \mathcal{M}$; see inset illustration. All adapters project to a shared output dimensionality

$D'$. Formally:

$$\mathcal{L} = \left\{ \mathrm{LA}_M : \mathbf{A}^p \mapsto \mathbf{A}^p \mathbf{W}_M \,\middle|\, \mathbf{W}_M \in \mathbb{R}^{D_M \times D'}, \, M \in \mathcal{M} \right\},$$

where $D_M$ is the hidden dimension of model $M$, and $\mathbf{A}^p$ denotes the pooled activation tensor. The result is a tensor of shape $L_p \times N_p \times D'$. This design—applying a single linear transformation, $\mathrm{LA}_M$, jointly across all layers and tokens of the activation tensor for a given LLM $M$, rather than using separate projections per layer or token—is motivated by the inherent alignment of feature spaces within the architecture of a single transformer model. Specifically, residual connections promote consistency across layers, while the weight sharing of the transformer across tokens ensures a uniform representation structure along the token dimension.

Resorting to a set of LLM-dependent LAs is particularly relevant in relation to principle (i). Indeed, this expedient ensures that the downstream architectural components process objects which are of consistent shape and are meaningfully comparable across the internal representation spaces of the distinct LLMs in $\mathcal{M}$. Indeed, motivated by recent literature on representation alignment [24, 45, 46, 32], we hypothesize that, despite differences in architectural design, training corpora, weight initializations and learning schemes, the hidden representations of LLMs can, in fact, be put in an approximate correspondence by some linear transformations, which we aim to learn via the adapters in $\mathcal{L}$. This contributes to rendering our multi-LLM setup not only computationally possible, but also meaningful from a learning perspective and, as we show in the next sections, also beneficial in terms of generalization performance. For an illustrative motivating example of how a linear mapping can tackle misalignment in the representation space, we refer the interested reader to Appendix C, where we specifically consider those induced by neuron symmetries, as recently studied in [47, 3].

**ViT-B.** In view of the proposed analogy between ATs and images, the output of any LA module is then processed à la ViT [14]. First, it is rearranged into a sequence of non-overlapping activation patches of size $(p_H, p_W)$. If $L_p$ and $N_p$ are divisible by, resp., $p_H$ and $p_W$, this gives $H = L_p/p_H$ and $W = N_p/p_W$ patches. The resulting reshaped tensor is denoted: $\mathbf{A}_{\text{patches}} \in \mathbb{R}^{(H \cdot W) \times p_H \times p_W \times D'}$.

We augment each patch with a per-patch learnable Positional Encoding (PE), providing intra-patch ordering over (pooled) activation pixels. The Transformer's native PEs provide a global ordering over activation patches. Combined with our per-patch PEs, this allows reconstructing each pooled pixel's original position. Similarly to ViT, the enriched activation patches are flattened and passed through a shared linear layer. The resulting sequence is then processed by a stack of Transformer blocks. Overall, using a vision-like backbone aligns with the aforementioned analogy between ATs and images and, as we will experimentally show next, it reveals to be an effective inductive bias.

## 5 Experiments

**Setup.** We experimentally analyze various aspects of learning on ATs with `ACT-ViT`. We consider, in particular, two relevant setups: (a) in-domain and (b) out-of-domain.

The in-domain setup (a) is the setting commonly considered in HD literature: it assesses the generalization performance on data from a *known* LLM on *known* generative tasks, i.e., it assumes training-time access to annotated data for a target LLM and task of interest. In this standard scenario, we attempt answering the following questions: ($\mathbf{Q_{(a)}1}$) Does our method outperform standard methods, including probing classifiers? ($\mathbf{Q_{(a)}2}$) What is the contribution of multi-dataset training? ($\mathbf{Q_{(a)}3}$) Is the vision architectural inductive bias effective? ($\mathbf{Q_{(a)}4}$) How scalable is our approach?

Harder – but more compelling – is the out-of-domain setup (b). Here, we benchmark methods in cases where either the target dataset is unseen at training time, or, even more challenging, both the target dataset and the target LLM are unseen. We study, in particular, the performance of our approach when either no data or only a small amount is available for adaptation. Those correspond, resp., to zero-shot generalization, and to fine-tuning only the LA in our pipeline, keeping the other parameters frozen (see inset). These settings reflect relevant real-world use-cases

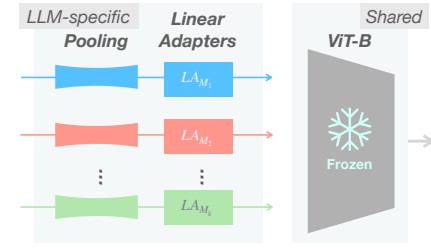

pertaining to, e.g., newly introduced LLMs and/or generative tasks with hardly accessible hallucination labels. Our experiments revolve around the following questions: **(Q_(b)1)** Can our method effectively generalize on unseen datasets? Does it adapt in a sample-efficient manner? **(Q_(b)2)** Can it handle unseen LLMs by only training the corresponding Linear Adapters? **(Q_(b)3)** Does this approach favorably compare with baselines?

**LLMs and datasets.**    Aligning with prior work, we focus on the datasets and LLMs considered in [48, 6]. We experiment with Mistral-7B-Instruct-v0.2 [27] (Mis-7B) and Llama-3-8B-Instruct [63] (LlaMa-8B) over the generation tasks encompassed by the following datasets: TriviaQA [28] (question answering), HotpotQA with (HQA-Wc) and without supporting context (HQA) [67] (question answering), IMDB movie review [40] (sentiment analysis), and Movies [48] (actor role retrieval). On top of this, we additionally include another LLM: Qwen2.5-7B-Instruct [66] (Qwen-7B). This setup yields a total of 15 distinct LLM-dataset combinations. We use the area under the ROC curve (AUC) to evaluate error detectors, a standard metric in this domain [48, 61, 6].

**Methods in comparison.**    We compare the performance of a diverse set of methods: **(1) Probabilities/logits-based.** To detect errors, several works [18, 29, 64, 23] assess LLM confidence by aggregating output token probabilities or logits by taking the mean, max, or min across these scores. We refer to these with the naming **Logit/Probas-mean/min/max**. These approaches operate in a less-restrictive setting, and thus constitute important baselines for our method. They are especially relevant in the out-of-domain setup (b), since, being *training-free*, these methods are (degenerately) sample-efficient by construction. We also compare against **LOS-Net** [6], a recent, *learnable* approach on output probabilities. **(2) Probing Classifiers** [7] constitute natural and effective white-box baselines for our approach [29, 36]. For each benchmark, we learn logistic regression probes on each layer of the LLM and across a fixed set of token positions $\mathcal{N}$ (see Section 4), a superset of the top-performing (static) tokens identified in [48]. The best layer-token combination, as well as the L2 regularization coefficient, is selected on the respective validation sets. We refer to this method as **Probe[∗]**. With **Token[n]**, $n \in \mathcal{N}$, we refer, instead, to the probe specific to token position $n$. **(3) Activation Tensors-Methods** are those designed and introduced in this work. `ACT-ViT`, (Section 4.2) is trained on the corpus of *all* available *training* sets across LLMs and datasets. In the out-of-domain setup (b), some of these are purposely segregated from the corpus in an effort to study domain adaptation abilities; this will be appropriately specified. We also benchmark two variants of our approach to separately study the benefits of multi-dataset training and architectural inductive biases. We use `ACT-ViT`(s) to refer to the single LLM-dataset counterpart of our method trained individually on each of the 15 LLM–dataset combinations. `ACT-MLP` and `ACT-MLP`(s) denote, instead, simplified variants in which MLPs operate on the output of Pool, flattened. These are trained, resp., on the full training corpus[3] or separately on each single training set. Across all these architectures, if not specified otherwise, we use the fixed pooling hyperparameter of $(L_p, N_p) = (8, 100)$.

Next we present our main results, with additional experimental details and experiments in Appendix A.

### 5.1   In-domain Hallucination Detection

**Comparing detection methods in-domain.**    Table 1 reports the performance of the aforementioned methods across all the 15 LLM–dataset combinations. The last row reports the gap between `ACT-ViT` and the best-performing baseline from prior work. First, our methods outperform both probability-based and probing classifiers on all settings, with the only exception being IMDB over Mis-7B. There, they closely follow the best scoring Probe[∗]. Additional comparison with LOS-Net is available in Table 7 in Appendix A.3. *These results yield a positive answer to (Q_(a)1), underscoring how leveraging the full activation tensor can significantly benefit automated HD.* Second, we note that `ACT-ViT` outperforms `ACT-ViT`(s) in 12 out of 15 cases, sometimes with a remarkable margin. These results demonstrate that joint training, as implemented in `ACT-ViT`, can outperform training on individual LLM–dataset combinations, as in `ACT-ViT`(s). The notable performance of `ACT-ViT` suggests that the underlying signal for HD is shared across different LLMs and datasets, and that a simple linear adapter (module LA) can already effectively bridge these. *Answering (Q_(a)2), we conclude that training on Activation Tensors sourced from diverse datasets and LLMs holds significant potential, and that our method can effectively harness it.* Last, we intriguingly note how the `ACT-MLP`

---

[3]With appropriate padding to $D_{\max}$ – the largest hidden size among the considered LLMs

Table 1: AUC HD performance comparison on all 15 LLMs-Datasets combinations. The best test AUC score is indicated in **Bold**, and the second-best is underlined. The last row reports the improvement achieved by `ACT-ViT` over the best-performing baseline from prior work.

| | Mis-7B | | | | | LlaMa-8B | | | | | Qwen-7B | | | | |
|---|---|---|---|---|---|---|---|---|---|---|---|---|---|---|---|
| | HQA | Movies | HQA-Wc | IMDB | TriviaQA | HQA | Movies | HQA-Wc | IMDB | TriviaQA | HQA | Movies | HQA-Wc | IMDB | TriviaQA |
| **Probas-based** | | | | | | | | | | | | | | | |
| Logits-mean | 61.00 | 63.00 | 55.00 | 57.00 | 60.00 | 65.00 | 75.00 | 56.00 | 59.00 | 66.00 | 66.20 | 71.30 | 67.40 | 74.80 | 68.20 |
| Logits-max | 53.00 | 54.00 | 51.00 | 47.00 | 54.00 | 59.00 | 67.00 | 56.00 | 51.00 | 54.00 | 60.40 | 65.10 | 60.60 | 60.70 | 63.90 |
| Logits-min | 61.00 | 66.00 | 53.00 | 52.00 | 63.00 | 67.00 | 71.00 | 55.00 | 55.00 | 74.00 | 59.80 | 42.10 | 58.50 | 72.10 | 61.60 |
| probas-mean | 63.00 | 61.00 | 56.00 | 54.00 | 60.00 | 61.00 | 73.00 | 56.00 | 73.00 | 67.00 | 67.50 | 74.20 | 66.00 | 74.60 | 69.70 |
| probas-max | 50.00 | 51.00 | 53.00 | 48.00 | 50.00 | 56.00 | 64.00 | 53.00 | 49.00 | 54.00 | 61.80 | 72.90 | 59.00 | 50.10 | 64.30 |
| probas-min | 58.00 | 60.00 | 52.00 | 51.00 | 59.00 | 60.00 | 65.00 | 51.00 | 57.00 | 67.00 | 54.40 | 44.70 | 53.60 | 65.40 | 57.40 |
| **Learning-based** | | | | | | | | | | | | | | | |
| Token [0] | 82.69 | 75.59 | 65.79 | **97.61** | 79.95 | 79.00 | 75.84 | 62.22 | 93.58 | 80.13 | 80.43 | 86.73 | 64.83 | 81.84 | 75.42 |
| Token [1] | 80.63 | 75.02 | 61.03 | 96.95 | 77.29 | 79.71 | 77.70 | 64.21 | 89.88 | 79.81 | 79.23 | 86.47 | 64.21 | 80.99 | 77.24 |
| Token [2] | 79.88 | 75.46 | 61.59 | 97.46 | 76.76 | 79.60 | 77.73 | 66.24 | 90.51 | 79.90 | 79.14 | 87.98 | 67.62 | 89.11 | 79.77 |
| Token [-3] | 74.19 | 69.17 | 61.34 | 92.91 | 74.26 | 70.73 | 75.76 | 63.11 | 82.26 | 76.06 | 79.29 | 88.14 | 71.98 | 90.87 | 84.12 |
| Token [-2] | 76.06 | 70.96 | 63.84 | 94.54 | 75.47 | 75.11 | 78.85 | 66.95 | 81.93 | 78.18 | 78.74 | 90.44 | 73.76 | 92.54 | 81.93 |
| Token [-1] | 77.89 | 74.87 | 66.80 | 94.99 | 76.82 | 77.98 | 83.61 | 70.56 | 80.49 | 81.09 | 80.57 | 93.22 | 75.01 | 93.76 | 87.08 |
| Probe[*] | 79.88 | 74.87 | 66.80 | **97.61** | 79.95 | 79.71 | 83.61 | 70.56 | 93.58 | 81.09 | 80.57 | 93.22 | 75.01 | 93.76 | 87.08 |
| ACT-MLP(s) | 78.76 | 70.64 | 59.51 | 93.65 | 78.28 | 77.95 | 76.53 | 65.18 | 91.22 | 78.23 | 82.57 | 87.47 | 72.57 | 89.07 | 86.10 |
| ACT-MLP | 78.41 | 74.25 | 60.57 | 94.79 | 78.11 | 77.77 | 76.33 | 63.26 | 90.23 | 78.61 | 82.70 | 87.46 | 72.75 | 90.60 | 85.53 |
| ACT-ViT(s) | 83.62 | 78.79 | 65.80 | 97.59 | 83.37 | 81.27 | 83.23 | 69.97 | 93.71 | 84.26 | **88.03** | 94.59 | 76.44 | 94.71 | **91.01** |
| ACT-ViT | **84.33** | **79.63** | **70.23** | 97.03 | **84.28** | **82.73** | **84.81** | **72.30** | **94.12** | **85.58** | 87.62 | **95.08** | **78.26** | **96.22** | **91.01** |
| Improvement | +1.64 | +4.04 | +3.43 | -0.58 | +4.33 | +3.02 | +1.20 | +1.74 | +0.54 | +4.49 | +7.05 | +1.86 | +3.25 | +2.46 | +3.93 |

baselines are often outperformed by Probe[∗]. *This clearly answers ($Q_{(a)}$3) and shed light on the pivotal role of the design choice behind `ACT-ViT`.*

**Run-time and performance trade-offs.** The inference run-time for `ACT-ViT` and `ACT-ViT(s)`, is extremely contained, around $10^{-5}$ seconds. This is orders of magnitude faster than techniques resorting to LLM-querying [33, 48]. The training time of `ACT-ViT` on the full training corpus (15 LLM-dataset combinations) is below three hours on a single NVIDIA L-40 GPU. Our competitive `ACT-ViT(s)` trains, instead, in $\approx$10 minutes. For detailed run-times see Table 9 in Appendix A.4.

One may wonder if it is possible to further reduce the run-time of our method with a more aggressive Pool. We study this by testing different configurations on Qwen-7B over HotpotQA. We train `ACT-ViT(s)` varying the number of pooled layers and tokens: $(L_p, N_p) \in \{(4, 20), (4, 100), (8, 20), (8, 100)\}$. Results, shown in Figure 3, are reported in terms of Test AUC (left) scores and corresponding training run-times (right). As expected, increasing $(L_p, N_p)$ improves performance, but at the cost of longer training. Notably, however, configuration $(L_p, N_p) = (4, 20)$ trains in around one minute while still achieving an AUC of 86%, $\approx$6 units higher than Probe[∗]. *In respect to ($Q_{(a)}$4), we conclude that our approach is indeed very scalable, and that operating on simple design choices can provide optimal performance-complexity trade-offs.*

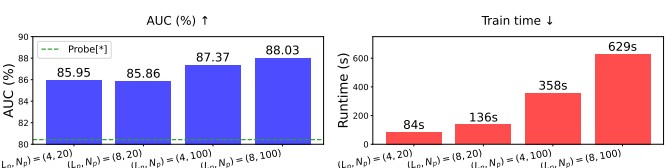

Figure 3: Ablation study on the pooling hyperparams $(L_p, N_p)$. Probe[∗] AUC is indicated by dashed green line in the left plot.

## 5.2 Off-domain Hallucination Detection

**Hallucination Detection on unseen datasets.** We implement a "leave-one-dataset-out" setup: for each LLM-dataset pairing, we train `ACT-ViT` on the corpus of the remaining 14 out of 15 combinations, and then evaluate on the target held-out test set zero-shot[4]. This is something fundamentally not possible with standard probing methods and hence compare against the best probability-based baseline, denoted **Best-Probas**, due to its training-free nature, and against `ACT-MLP`.

Results are shown as bar plots in Figure 4 on all 15 LLM–dataset combinations. We observe strong generalization performance: `ACT-ViT` outperforms Best-Probas in 13 out of 15 cases, sometimes substantially. On the IMDB dataset with Mis-7B, `ACT-ViT` achieves a gain of +37 AUC points.

---

[4]On held-out LLM-data pair $(M, \mathcal{D})$, this is possible as $M$'s LA is pretrained on the pairs $(M, \mathcal{D}')$, $\mathcal{D}' \neq \mathcal{D}$.

`ACT-MLP` also shows reasonable performance, but it consistently falls short of `ACT-ViT`. *These results provide a positive answer to ($Q_{(b)}1$).*

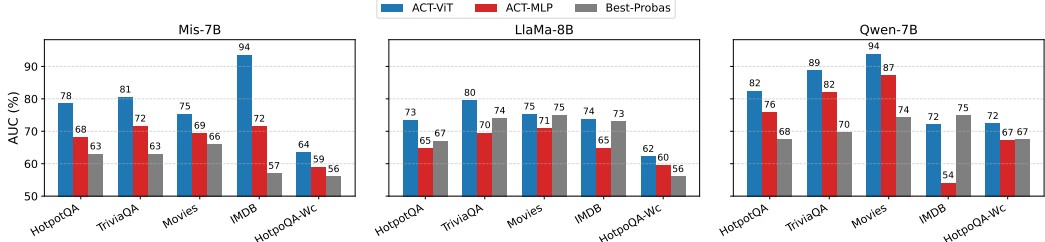

Figure 4: Zero-shot generalization results across all 15 LLM–dataset combinations, in a "leave-one-dataset-out" setup. Each bar shows the Test AUC score of `ACT-ViT`, `ACT-MLP`, and the best probability-based baseline – Best-Probas – on a dataset they were not trained on.

We next consider a more tolerant setting whereby only a limited amount of samples from the target training set is available. In particular, we start with the pre-trained `ACT-ViT` and fine-tune the LA module keeping the ViT-B component frozen. We compare with Best-Probas and Probe[∗], fitted only the corresponding available training data. Figure 5 shows Test AUCs for Mis-7B over HotpotQA across varying fine-tuning set sizes; results for other 15 LLM-dataset combinations are in Appendix A.6. `ACT-ViT` consistently outperforms all baselines across all training fractions. Importantly, the performance gap between `ACT-ViT` and Probe[∗] widens as the training set shrinks, indicating that the frozen ViT-B backbone captures strong, transferable features learned from (pre)training on other LLMs and datasets. Additionally, `ACT-ViT` requires only 10% of the available training data (1k samples) to surpass Probe[∗] when trained on the full 10k-sized training set.

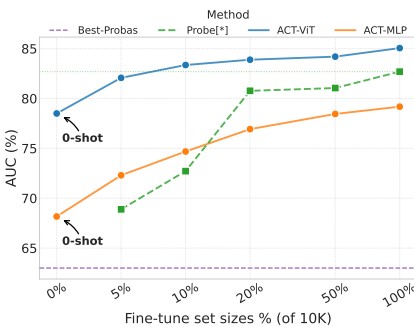

Figure 5: Low-data regime results, for Mis-7B over the HotpotQA dataset.

**Hallucination Detection on unseen LLMs and datasets.** We finally benchmark our method in the following challenging setup: we train `ACT-ViT` on 2 out of 3 LLMs and leave the third one out. Specifically, our model is pretrained on a corpus of 10 out of 15 training sets containing no data from target LLM $M$. It is then (separately) adapted on the remaining 5 datasets by keeping ViT-B frozen and only training the $LA_M$ module from scratch. In Table 2 we report the obtained Test AUCs, comparing `ACT-ViT` against Best-Probas and Probe[∗]. `ACT-ViT` outperforms these baselines in 13 out of 15 cases. Recalling that the ViT-B module is frozen during adaptation, this further demonstrates how the features learned during (pre)training over different LLMs can strongly transfer to an unseen one. *Along with the above, these results answer positively to $Q_{(b)}2$ and $Q_{(b)}3$.*

Table 2: AUC comparison with only the LA fine-tuned (ViT-B frozen) on an out-of-domain LLM and dataset. Last row shows `ACT-ViT`'s gain over the best baseline from prior work.

|  | **Mis-7B** | | | | | **LlaMa-8B** | | | | | **Qwen-7B** | | | | |
| --- | --- | --- | --- | --- | --- | --- | --- | --- | --- | --- | --- | --- | --- | --- | --- |
|  | HotpotQA | Movies | HQA-Wc | IMDB | TriviaQA | HotpotQA | Movies | HQA-Wc | IMDB | TriviaQA | HotpotQA | Movies | HQA-Wc | IMDB | TriviaQA |
| Best-Probas | 63.00 | 66.00 | 56.00 | 57.00 | 63.00 | 67.00 | 75.00 | 56.00 | 73.00 | 74.00 | 50.00 | 50.00 | 50.00 | 50.00 | 50.00 |
| Probe[∗] | 82.69 | 74.87 | 66.80 | **97.61** | 79.95 | 79.71 | **83.61** | 70.56 | 93.58 | 80.13 | 80.57 | 93.22 | 75.01 | 93.76 | 87.08 |
| `ACT-MLP` | 80.22 | 72.83 | 60.68 | 95.05 | 79.64 | 76.83 | 75.51 | 64.50 | 88.30 | 76.65 | 83.40 | 86.03 | 73.64 | 89.14 | 86.34 |
| `ACT-ViT` | **83.73** | **79.07** | **68.87** | 97.58 | **83.04** | **80.83** | 82.31 | **70.66** | **94.71** | **82.63** | **87.27** | **95.00** | **77.56** | **96.25** | **91.14** |
| Improvement | +1.04 | +4.20 | +2.07 | −0.03 | +3.09 | +1.12 | −1.30 | +0.10 | +1.13 | +2.50 | +6.70 | +1.78 | +2.55 | +2.49 | +4.06 |

# 6   Conclusions

We introduce `ACT-ViT`, a fast and effective method for LLM Hallucination Detection (HD) that relies solely on a model's internal representation of a single response. Unlike traditional probing,

`ACT-ViT` leverages the full *Activation Tensor* (AT) and, drawing on an analogy with images, it applies techniques from modern vision architectures, enabling both cross-LLM training and generalization. It is dramatically faster than multi-query methods (runtime $\approx 10^{-5}$ seconds), making it suitable for real-time use. Across 15 LLM–dataset pairs, `ACT-ViT` consistently outperforms prior white-box methods, supports zero-shot generalization to new datasets, and adapts efficiently to unseen LLMs. Our work opens several directions for future research. While focused on HD, `ACT-ViT` can be extended to tasks like data contamination and LLM-generated content detection. More advanced architectures also warrant exploration, e.g., replacing per-LLM adapters with a shared module that exploits permutation symmetries across ATs.

**Limitations.** Due to the large size of the ATs, we begin their processing by conducting a simple pooling operation to reduce their dimensionality. While effective in managing computational costs, this step may discard potentially informative signals. Developing more sophisticated approaches to handle AT size without sacrificing signal fidelity remains a promising future direction.

## Ackowledgements

G.B. is supported by the Jacobs Qualcomm PhD Fellowship. F.F. conducted this work supported by an Aly Kaufman Post-Doctoral Fellowship. H.M. is a Robert J. Shillman Fellow and is supported by the Israel Science Foundation through a personal grant (ISF 264/23) and an equipment grant (ISF 532/23). F.F. is extremely grateful to the members of the "Eva Project", whose support he immensely appreciates.

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

# A  Extended Experimental Section

Our experiments were conducted using the PyTorch [50] framework (License: BSD), using a single NVIDIA L-40 GPU for all experiments. We use a fixed batch size of 128 for all experiments, other than the ones with `ACT-ViT(s)`, `ACT-MLP(s)`, where we used a batch size of 64. We used 4 heads in the transformer part of ViT for all experiments. Hyperparameter tuning was performed utilizing the Weight and Biases framework [9] – see Appendix A.1.

**Optimizer and Schedulers.** For all datasets, we use the AdamW optimizer [39] in combination with a cosine learning rate scheduler, incorporating a warm-up phase over the first 10% of training epochs.

**LLMs.** We consider the following LLMs for our experiments:

1. **Mistral-7b-instruct-v0.2** [27] (License: Apache-2.0). Referred to as Mis-7B in the main text and accessed through the Hugging Face interface at `https://huggingface.co/mistralai/Mistral-7B-Instruct-v0.3`.

2. **Llama-3-8b-Instruct** [63] (License: Llama-3[5]). Referred to as LlaMa-8B in the main text and accessed through the Hugging Face interface at `https://huggingface.co/meta-llama/Meta-Llama-3-8B-Instruct3`.

3. **Qwen-2.5-7b-Instruct** (License: Apache-2.0): Referred to as Qwen-7B in the main text and accessed through the Hugging Face interface at `https://huggingface.co/Qwen/Qwen2.5-7B-Instruct`.

**Generating Activation Tensors.** Given a large language model (LLM) $M$ and an output sequence of $N$ tokens, we define the *activation tensor* $\mathbf{A} \in \mathbb{R}^{L_M \times N \times D_M}$, where $L_M$ is the number of layers in $M$, and $D_M$ is the hidden dimension. Let $\mathbf{A}^{(l)} := \mathbf{A}[l, :, :]$ denote the hidden representations at the $l$-th layer across the entire sequence. In a standard transformer architecture, these representations are computed as:

$$\mathbf{A}^{(l)} = \mathbf{A}^{(l-1)} + \left[ \text{ReLU}\left( \texttt{Attn}(\mathbf{A}^{(l-1)})\mathbf{W}_l^1 + \mathbf{b}_l^1 \right) \right] \mathbf{W}_l^2 + \mathbf{b}_l^2, \tag{1}$$

where `Attn` denotes the output of the multi-head attention mechanism, followed by batch normalization. The matrices $\mathbf{W}_l^1, \mathbf{W}_l^2 \in \mathbb{R}^{D_M \times D_M}$ and biases $\mathbf{b}_l^1, \mathbf{b}_l^2 \in \mathbb{R}^{D_M}$ correspond to the feed-forward subnetwork at layer $l$. Throughout this paper, we use the term *activation tensor* to refer to $\mathbf{A}$ as defined in Equation (1).

## A.1  HyperParameters

This section outlines the hyperparameter search performed for our experiments. We employ the same hyperparameter grid for both our primary model, `ACT-ViT`, and our proposed baseline, `ACT-MLP`[6]. Hyperparameter grid search is performed to optimize the AUC on the validation set. The best hyperparameters are selected based on the run that achieves the highest validation AUC, for each of our two proposed methods, namely `ACT-ViT`, and `ACT-MLP`. The hyperparameter grids used for all experiments are outlined below, corresponding to each experimental setup:

1. Training on a single LLM-specific dataset – see Table 3.
2. Joint training on all 15 datasets simultaneously – see Table 4.
3. Leave-one-out training (excluding 1 out of 15 datasets) – upper part of Table 5.
4. Low-data regime adaptation on the held-out dataset – lower part of Table 5. "–" refers to slot which are inapplicable, or taken from the pre-trained model.
5. Leaving out one LLM (excluding 5 out of 15 LLM-dataset combinations) – see Table 6.

For the probing baselines—specifically `Token[n]` for $n \in \mathcal{N}$ (see Section 4) and `Probe[*]`—we performed a grid search over inverse regularization strengths $C \in \{10000, 100, 1.0, 0.01, 0.0001\}$. For each token probe, we selected the best-performing model based on validation set performance, corresponding to the optimal value of $C$. In the case of `Probe[*]`, we additionally selected the best token position using the validation set.

---

[5] `https://huggingface.co/meta-llama/Meta-Llama-3-8B/blob/main/LICENSE`

[6] The patch size parameter does not apply to these baselines, as it is not part of their configuration.

Table 3: Hyperparameter search grid for `ACT-ViT`(s) and `ACT-MLP`(s), for each of the 15 LLM-dataset combinations.

| Hyperparameter Search Grid for each of the 15 LLM-dataset combinations | |
|---|---|
| Number of layers | $\{1, 3\}$ |
| Learning rate | 0.001 |
| Embedding size | $\{128, 1024\}$ |
| Epochs | 15 |
| Dropout | 0.3 |
| Weight Decay | $\{1, 0.001\}$ |
| Patch Size | $\{(1, 1), (8, 1), (4, 2)\}$ |

Table 4: Hyperparameter search grid for `ACT-ViT` and `ACT-MLP`, in the setting where training is performed jointly across all 15 LLM–dataset combinations.

| Hyperparameter Search Grid for joint-training of the 15 LLM-dataset combinations | |
|---|---|
| Number of layers | 3 |
| Learning rate | $\{0.001, 0.0005\}$ |
| Embedding size | 128 |
| Epochs | 5 |
| Dropout | 0.3 |
| Weight Decay | $\{10, 0.001\}$ |
| Patch Size | $\{(1, 1), (1, 2), (1, 4), (2, 1), (4, 1)\}$ |

Table 5: Hyperparameter search grid used for `ACT-ViT` and `ACT-MLP` when trained on the combined set of all 14 LLM datasets, in the "leaving one dataset out" setup. "–" denotes slots that are inapplicable, as their values are inherited from the pre-trained model.

| Num. layers | Learning rate | Embedding size | Epochs | Dropout | Weight Decay | Patch size |
|---|---|---|---|---|---|---|
| **Pre training (and Zero-shot)** | | | | | | |
| 3 | 0.001 | 128 | 15 | 0.3 | 0.001 | $(1, 1)$ |
| **Low-data regime LA adaptation (over $\{5\%, 10\%, 20\%, 50\%, 100\%\}$ of 10,000 test samples)** | | | | | | |
| – | 0.001 | – | 5 | – | 0.001 | – |

Table 6: Hyperparameter search grid used for `ACT-ViT` and `ACT-MLP` when trained on the combined set of 10 LLM datasets, in the "leaving one LLM out" setup. "–" denotes slots that are inapplicable, as their values are inherited from the pre-trained model.

| Num. layers | Learning rate | Embedding size | Epochs | Dropout | Weight Decay | Patch size |
|---|---|---|---|---|---|---|
| **Pre training** | | | | | | |
| $\{3, 5\}$ | 0.001 | 128 | 5 | 0.3 | 0.001 | $\{(1, 1), (8, 1), (4, 2)\}$ |
| **LA Adaptation** | | | | | | |
| – | 0.001 | – | 15 | – | $\{0.1, 0.01, 10, 20\}$ | – |

## A.2 Dataset Description

In this section, we provide an overview of the five datasets used in our analysis; we mostly follow the framework given in [48] in constructing the datasets. We aimed to cover diverse tasks, reasoning skills, and datasets, highlighting each one's unique value and how it complements the rest.

For all datasets, we used a consistent split of 10,000 training samples and 10,000 test samples, unless otherwise specified. From the 10,000 training samples, 20% (i.e., 2,000) were selected in a stratified manner for validation, using a fixed random seed of 42.

1. **HotpotQA with and without context** (License: CC-BY-SA-4.0) [67]: HotpotQA is a multi-hop question answering dataset featuring diverse questions that require reasoning across multiple sources. Each instance includes supporting Wikipedia documents. We use two settings in our analysis: (1) *Without context*, where questions are presented alone, testing factual recall and reasoning; and (2) *With context*, where supporting documents are provided, emphasizing the model's ability to leverage contextual information effectively.

2. **Movies** [48] (License: MIT): We use this dataset to evaluate generalization in scenarios regarding movies involving factual inaccuracies (i.e., hallucinations). This dataset contains 7857 test samples.

3. **IMDB** (originally released with no known license by Maas et al. [40]): This dataset consists of movie reviews for sentiment classification. Following the method in [48], we employed a one-shot prompt to help the large language model (LLM) apply the predefined sentiment labels accurately.

4. **TriviaQA** (originally released with no known license by [28]): A dataset of trivia question-answer pairs presented to the LLM without any supporting context, relying only on its internal parametric knowledge. Multiple acceptable answer variants are provided to facilitate automatic evaluation of response accuracy.

## A.3 Additional Results

Below in Table 7 we provide a comparison of `ACT-ViT` with LOS-Net.

Table 7: Comparison of `ACT-ViT` with LOS-Net. For LOS-Net, we report the 1-sigma error-bars. Best is in **Bold**.

| Method | HotpotQA | IMDB | Movies | HotpotQA | IMDB | Movies | HotpotQA | IMDB | Movies |
|---|---|---|---|---|---|---|---|---|---|
| | | Mis-7B | | | LLaMa-8B | | | Qwen-7B | |
| LOS-Net | $72.92 \pm 0.45$ | $94.73 \pm 0.58$ | $72.20 \pm 0.66$ | $72.60 \pm 0.34$ | $90.57 \pm 0.28$ | $77.43 \pm 0.66$ | $73.71 \pm 1.21$ | $88.19 \pm 0.88$ | $88.01 \pm 0.39$ |
| ACT-ViT | **84.33** | **97.03** | **79.63** | **82.73** | **94.12** | **84.81** | **87.62** | **96.22** | **95.08** |

`ACT-ViT` **performance average over three seeds.** As outlined in the NeurIPS paper checklist, we provide additional results for `ACT-ViT` over the datasets and tasks provided Table 1, where we rerun `ACT-ViT` with three random seeds and report the mean and standard deviation. Those results are provided in Table 8.

Table 8: `ACT-ViT` AUC HD performance averaged over three seeds with 1-sigma error bars.

| | HQA | Movies | HQA-Wc | IMDB | TriviaQA |
|---|---|---|---|---|---|
| Mis-7B | $84.48 \pm 0.28$ | $79.64 \pm 0.25$ | $69.87 \pm 0.63$ | $97.09 \pm 0.17$ | $84.18 \pm 0.57$ |
| LlaMa-8B | $82.45 \pm 0.36$ | $84.05 \pm 0.58$ | $71.82 \pm 0.34$ | $93.57 \pm 0.34$ | $85.25 \pm 0.35$ |
| Qwen-7B | $88.03 \pm 0.32$ | $95.25 \pm 0.14$ | $78.23 \pm 0.59$ | $96.24 \pm 0.05$ | $91.12 \pm 0.10$ |

## A.4 Run-Time

Table 9: Training time (in minutes [m] and seconds [s]) for Probe[∗] and `ACT-ViT` across all 15 LLM–dataset combinations. Each `ACT-ViT` training run was performed on a single NVIDIA L-40 GPU.

| LLM | Method | HotpotQA | Movies | HQA-Wc | IMDB | TriviaQA |
|---|---|---|---|---|---|---|
| **Mis-7B** | Probe[∗] | 2[m] 45[s] | 6[m] 23[s] | 1[m] 41[s] | 1[m] 38[s] | 1[m] 50[s] |
| | ACT-ViT | 13[m] 2[s] | 11[m] 28[s] | 12[m] 49[s] | 14[m] 17[s] | 12[m] 42[s] |
| **LlaMa-8B** | Probe[∗] | 2[m] 50[s] | 2[m] 5[s] | 1[m] 54[s] | 2[m] 22[s] | 2[m] 36[s] |
| | ACT-ViT | 13[m] 48[s] | 27[m] 5[s] | 16[m] 34[s] | 13[m] 1[s] | 27[m] 21[s] |
| **Qwen-7B** | Probe[∗] | 2[m] 0[s] | 1[m] 43[s] | 1[m] 51[s] | 1[m] 39[s] | 1[m] 53[s] |
| | ACT-ViT | 10[m] 29[s] | 13[m] 12[s] | 11[m] 49[s] | 11[m] 30[s] | 10[m] 46[s] |

We present the run-time of `ACT-ViT`(s), on a single NVIDIA L-40 GPU, compared with probing classifiers, on each of the 15 LLM-dataset combinations considered in the paper; see Table 9.

### A.5 Layer/Token Visualizations

Below we present the layer-token heatmaps, corresponding to each of the 15 LLM-dataset combination considered in our paper.

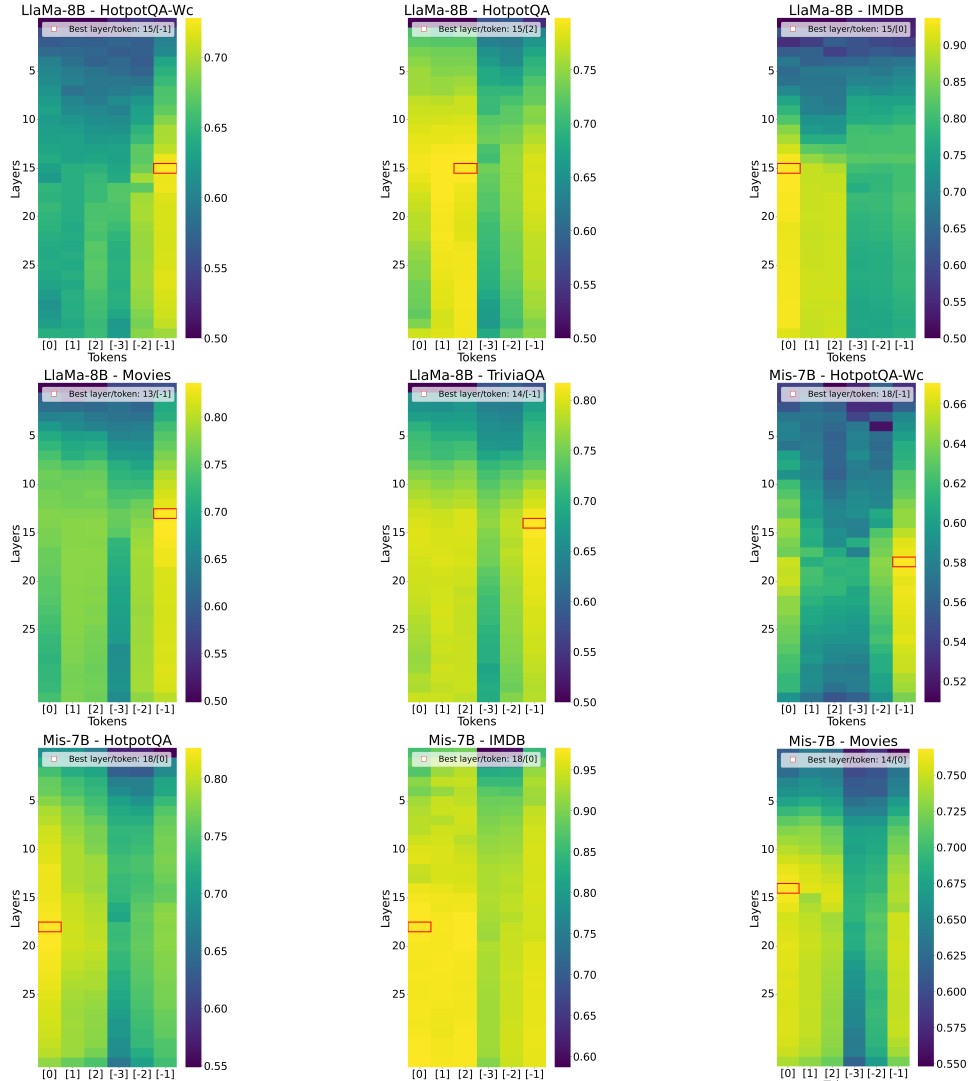

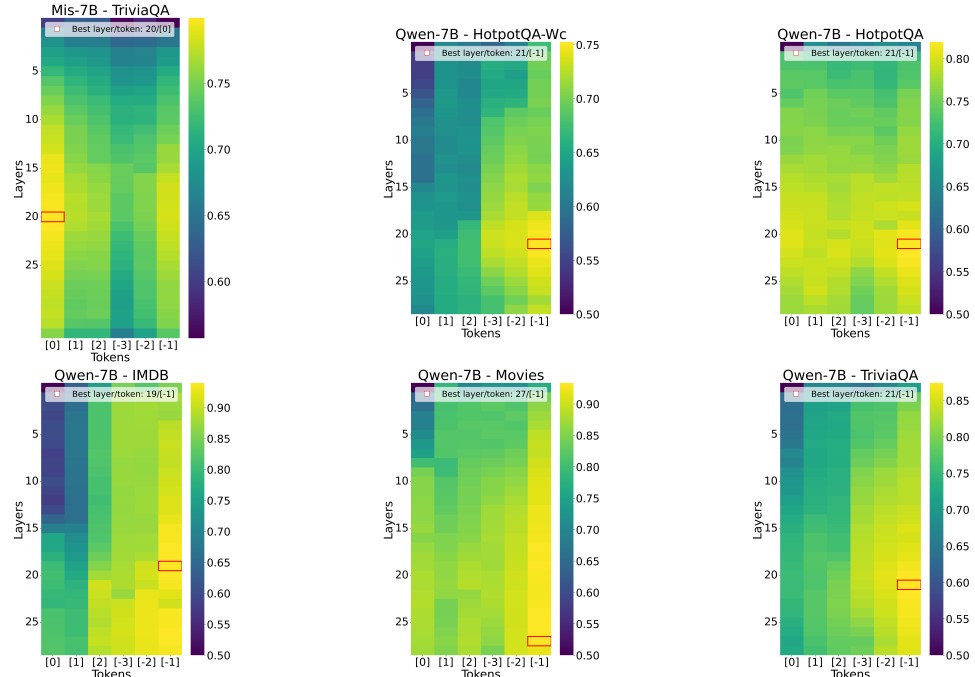

## A.6 Low-Data Regime Adaptation

Below, we present additional results in the low-data training regime for all 15 LLM–dataset combinations considered in our paper.

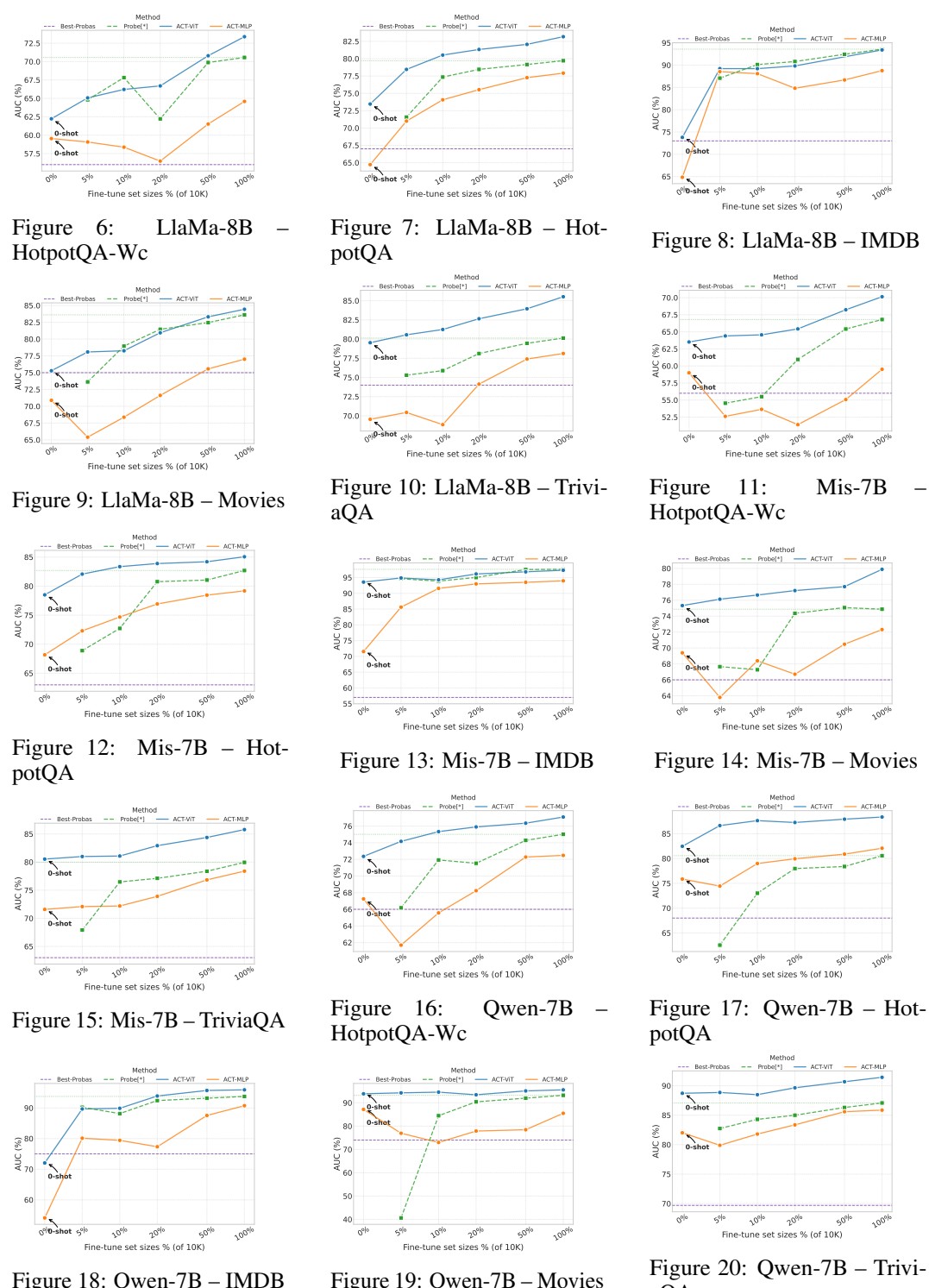

Figure 6: LlaMa-8B – HotpotQA-Wc

Figure 7: LlaMa-8B – HotpotQA

Figure 8: LlaMa-8B – IMDB

Figure 9: LlaMa-8B – Movies

Figure 10: LlaMa-8B – TriviaQA

Figure 11: Mis-7B – HotpotQA-Wc

Figure 12: Mis-7B – HotpotQA

Figure 13: Mis-7B – IMDB

Figure 14: Mis-7B – Movies

Figure 15: Mis-7B – TriviaQA

Figure 16: Qwen-7B – HotpotQA-Wc

Figure 17: Qwen-7B – HotpotQA

Figure 18: Qwen-7B – IMDB

Figure 19: Qwen-7B – Movies

Figure 20: Qwen-7B – TriviaQA

# B  Pooling Algorithm

Our pooling algorithm is presented in Algorithm 1.

**Algorithm 1** Pooling

---

**Require:** Tensor $\mathbf{A} \in \mathbb{R}^{L_M \times N \times D_M}$, target sizes $L_p$, $N_p$
**Ensure:** Tensor $\mathbf{A} \in \mathbb{R}^{L_p \times N_p \times D_M}$
1: $\mathbf{A} \leftarrow \texttt{permute}(\mathbf{A}, [D_M, L_M, N])$
2: Pad $\mathbf{A}$ so that $L_M$ and $N$ are divisible by $L_p$ and $N_p$
3: Compute patch sizes: $f_L = \frac{L_{\text{pad}}}{L_p}$, $f_N = \frac{N_{\text{pad}}}{N_p}$
4: Apply pooling:
$$\mathbf{A} \leftarrow \texttt{F.max\_pool2d}(\mathbf{A}, \texttt{kernel\_size} = (f_L, f_N))$$
5: $\mathbf{A} \leftarrow \texttt{permute}(\mathbf{A}, [L_p, N_p, D_M])$
6: **return A**

---

## C  Permutation Symmetries

Given the same number of layers and hidden dimensions, two LLMs can compute the exact same function and yet produce entirely different activation tensors. This discrepancy stems from internal symmetries in the model's weight space. To illustrate this, we consider a simplified transformer architecture with $L$ layers, omitting residual connections and batch normalization for clarity (though the analysis extends to those cases as well).

Recall that $\mathbf{A}^l$ is the activation tensor at layer $l \in [L]$. Let $(\mathbf{Q}_l, \mathbf{K}_l, \mathbf{V}_l)$ denote the query, key, and value matrices at layer $l$, computed as:

$$(\mathbf{Q}_l, \mathbf{K}_l, \mathbf{V}_l) = (\mathbf{A}^{l-1}\mathbf{W}_l^{Q_l}, \mathbf{A}^{l-1}\mathbf{W}_l^{K_l}, \mathbf{A}^{l-1}\mathbf{W}_l^{V_l}). \tag{2}$$

Assume the feed-forward network (FFN) at layer $l$ produces output:

$$\mathbf{A}^l = \text{ReLU}(\texttt{Attn}(\mathbf{A}^{l-1})\mathbf{W}_l^1 + \mathbf{b}_l^1)\mathbf{W}_l^2 + \mathbf{b}_l^2. \tag{3}$$

Now assume the model uses $d$-dimensional hidden features, and let $P \in \mathbb{R}^{d \times d}$ be a permutation matrix, and define the following weight transformation:

$$(\mathbf{W}_l^2, \mathbf{b}_l^2) \rightarrow (\mathbf{W}_l^2 P^\top, \mathbf{b}_l^2 P^\top), \tag{4}$$

$$(\mathbf{W}_{l+1}^{Q_{l+1}}, \mathbf{W}_{l+1}^{K_{l+1}}, \mathbf{W}_{l+1}^{V_{l+1}}) \rightarrow (P\mathbf{W}_{l+1}^{Q_{l+1}}, P\mathbf{W}_{l+1}^{K_{l+1}}, P\mathbf{W}_{l+1}^{V_{l+1}}). \tag{5}$$

Although the new weights (the right hand side) are very different, it is straightforward to verify that the permutation matrices cancel out, leaving the underlying function unchanged. However, recalling Equation (3), the activation tensors differ significantly, as the transformation permutes the feature dimension as follows:

$$\mathbf{A}[l, n, d] \rightarrow \mathbf{A}[l, n, \sigma(d)], \tag{6}$$

where $\sigma$ is the permutation induced by $P$.

This illustrative example remarks the relevance of resorting to LLM-specific LA modules, as opposed to using a single shared linear adapter. Although padding can side-step the dimensionality problem, such a single adapter would be required to implicitly learn (to be invariant to) all feature permutations that may potentially arise ($D!$). While we note that principled approaches for handling such symmetries exist – such as designing layers that are invariant to feature permutations by design – we consider these less relevant to us given the limited number of considered LLMs. Exploring symmetry-aware architectures still remains a promising direction we envision to explore in future endeavors.

**Extended Analysis to Standard Transformers Activations.** The implications extend naturally to a standard transformer that includes residual connections and batch normalization. Residual connections do not affect the analysis, since in our analysis the permutation symmetry is applies uniformly across layers—the same permutation is used for all weight matrices, so the residual addition remains consistent with our analysis. Batch normalization also preserves this symmetry, as it is a permutation-equivariant operation: applying a permutation to the input features results in a correspondingly permuted output.

# D Broader Impact

By enhancing the detection of hallucinations in Large Language Models, our work contributes to the responsible development and deployment of generative AI by promoting greater transparency and trust. However, we acknowledge that our findings also reveal aspects of the predictive information embedded within LLM internals. This insight could be misused by malicious actors, potentially motivating restrictions on LLM internals access and thereby slowing progress in open research.

