# OpenReview forum: "Beyond Token Probes: Hallucination Detection via Activation Tensors with ACT-ViT"
_NeurIPS.cc/2025/Conference — NeurIPS 2025 poster_

### Official Review · Reviewer_kwZz · 2025-06-17

**Clarity:** 2
**Significance:** 2
**Originality:** 3
**Rating:** 4
**Confidence:** 1

**Summary:**

This paper presents an LLM hallucination detection approach.
The motivation is that previous probing layer methods are largely limited by their generalization capability to new tasks, new datasets, and new models.
The key idea of the proposed method is to leverage the activation of LLMs.
After the initial extraction of several LLMs with respect to multiple queries, the authors then utilize a ViT model to capture some spatial relationship between responses and LLM layers.
Across 15 LLM-dataset pairs, ACT-ViT consistently outperforms prior white-box models and supports zero-shot generalization to new datasets.

**Questions:**

I will update my score if the authors can well address my concerns after rebuttal.

**Ethical Concerns:**

["NO or VERY MINOR ethics concerns only"]

**Final Justification:**

Though I still hold concerns with the necessity of this task, I agree with other reviewers that this work has some merits.

**Quality:**

2

**Strengths And Weaknesses:**

Strengths:
- The paper presents an interesting ensemble method to detect LLM hallucination, which may be useful for future research.
- The proposed method can outperform the previous probing layer baselines by a large margin.
- Across 15 LLM-dataset pairs, ACT-ViT consistently outperforms prior white-box models and supports zero-shot generalization to new datasets.

Weaknesses:
- I don't quite grasp the usefulness of this task. To me, the hallucinations for different LLMs are quite subtle and fine-grained, and may range from token-level to sentence-level. And the hallucination even is different for the same two input queries repeated. In other words, hallucination is dynamic, while detection is static.
- How did the authors get the labels for each response? Is it based on each token?
- It seems the pooling layers are carefully selected. This will limit the generalization of the proposed method.
- Are all the involved parameters trainable? If so, why use a ViT model?
- Following the previous concern, I don't see the necessity of using a ViT model for the ensemble processing.

Minor:
Line 111, their internal

---

> ### Author Rebuttal · Authors · 2025-07-30
>
> Thank you for your thoughtful feedback. We're glad you found our method interesting and valuable for future research. We also appreciate your recognition that ACT-ViT consistently outperforms prior approaches. Below, we address each of the points you raised.
>
> **"I don't quite grasp the usefulness of this task. To me, the hallucinations for different LLMs are quite subtle and fine-grained, and may range from token-level to sentence-level. And the hallucination even is different for the same two input queries repeated. In other words, hallucination is dynamic, while detection is static."**
>
> Thank you for the opportunity to clarify. ACT-ViT does not attempt to produce a single classification score shared across multiple LLMs. While it is trained jointly on activation tensors from different LLMs, the input to our model and the prediction is made specifically, per each LLM's response.
> Furthermore, as you rightly pointed out, hallucinations are nuanced and may manifest differently across LLMs and granularities. Their dynamic and context-dependent nature actually motivates our approach. Rather than using heuristics or model specific, static, single position probes, ACT-ViT learns from full activation tensors from diverse LLMs and datasets. This way it can extract predictive features at the level of tokens, sentences, or otherwise, within each tensor and learn from a diverse set of hallucination signals. By outperforming the aforementioned baselines (e.g., see Table 1 in the paper), we validate that this joint training enables ACT-ViT to identify relevant patterns within each activation tensor, effectively learning to adapt its focus to the dynamic nature of hallucination.
>
> **"How did the authors get the labels for each response? Is it based on each token?"**
>
> We followed the approach described in [1] to label the responses, performing the labeling at the sentence level rather than the token level. This involved matching sentences from the LLM-generated responses to those in the ground-truth answers using routines involving string matching.
>
> **"It seems the pooling layers are carefully selected. This will limit the generalization of the proposed method."**
>
> We would like to clarify our pooling strategy – the full pooling algorithm can be found in Appendix B, Algorithm 1. Our pooling technique selects the pooling kernel such that we end up with a fixed final size $(L_p, N_p)=(8, 100)$, which was fixed and kept constant across all experiments in the paper; except the ablation study in Figure 3, which specifically ablates this size.
> While the pooling does introduce some information loss, it proved effective in practice, as demonstrated by our results throughout the paper (e.g., see Table 1). Furthermore, we conducted an ablation study on different pooling sizes (see Figure 3), which shows a trade-off between performance and computational complexity: larger pooling dimensions $(L_p, N_p)$ reduce information loss and improve performance, but at the cost of increased training time. This flexibility allows our pooling strategy to be adapted based on the user’s computational constraints. Notably, as shown in Figure 3, all pooling variants continue to outperform the strongest probing baseline, demonstrating a degree of robustness across configurations.
>
> **"Are all the involved parameters trainable? If so, why use a ViT model?"**
>
> Thank you for your question. Our ViT backbone (ViT-B), is not pre-trained on vision tasks: in fact, all parameters in our pipeline are trained from scratch on a corpus of hallucination detection datasets; an overview of our method is available in Figure 1 in the paper. After this procedure, and only in the off-domain experiments, ViT-B is kept frozen and only the linear adapters are trained.
> In short, we do not use a pretrained ViT model, but rather adopt a ViT-like architecture we train from scratch for hallucination detection. We chose this architectural design pattern due to the structural analogy between Activation Tensors and images (two sequential dimensions, i.e., tokens and layers, arranged spatially in a 2D map). We elaborate on this analogy in our paper, in Section 4.1, lines 174–184. This motivates the use of a ViT-like architectural backbone, which proves to be effective in our experiments as it outperforms its MLP-based counterpart. In the next revision, we will make sure to highlight this rationale and aspects more clearly.
>
> **"Following the previous concern, I don't see the necessity of using a ViT model for the ensemble processing."**
>
> We believe the ViT backbone (ViT-B) plays an important role in our setup, and we would like to stress that it is not used for ensemble processing (e.g., combining outputs from multiple LLMs), but rather as a unified processing architecture on activation tensors. To clarify, the overall approach has two components: (1) we use pooling and linear adapters to bring activation tensors from different LLMs into a consistent shape and space, allowing joint training across models; and (2) this resulting tensor, structurally analogous to an image as explained in our previous responses, is then processed by ViT-B. As already mentioned, ViT-B is a natural and effective fit for this, as it allows to process activation tensors in a structured and spatially-aware way, beyond static, single position probing. We will make these rationales more explicit in the next revision.
>
> **"Minor: Line 111, their internal"**
>
> Thanks for the note, we will fix this in the next version of the paper.
>
> We appreciate your thoughtful feedback and hope our responses have addressed all of your points. If any issues remain unclear, we would be happy to provide further clarification. Otherwise, as you mentioned, we kindly ask you to consider updating your score based on our responses.
>
> **References:**
>
> [1] Llms know more than they show: On the intrinsic representation of llm hallucinations. Orgad et al., ICLR 2024

---

> > ### Comment · Reviewer_kwZz · 2025-08-03
> > **Response to rebuttal**
> >
> > The authors' responses partially addressed my concerns. I found my biggest concern lies in the necessity of this task, rather than the authors' implementations. Since I'm not that familiar with this task, I will try to calibrate with other reviewers who have more knowledge.
> >
> > In addition, I still feel like pooling layers that are carefully selected will limit the generalization of the proposed method.

---

> > > ### Author Response · Authors · 2025-08-07
> > >
> > > Thank you for your thoughtful comments. We appreciate your openness to calibrate your score with other reviewers. In our revision, we will more clearly articulate the task’s motivation and justify its necessity.

---

### Official Review · Reviewer_gx9D · 2025-06-27

**Clarity:** 4
**Significance:** 3
**Originality:** 3
**Rating:** 5
**Confidence:** 4

**Summary:**

This paper introduces a novel vision transformer-inspired model for probing LLM activations for the task of hallucination detection. They process all the activations across layers and token positions as input, showing this enables cross dataset and LLM (with LLM-specific adapters) generalization, and strong performance at hallucination detection.

**Questions:**

- This is framed as hallucination detection but it seems like the task is accuracy prediction. Could you clarify your definitions?
- Have you investigated other probing tasks?
- Have you looked into interpreting the ACT-ViT itself? What layers/tokens are relevant and whether they agree with the single token/layer results?
- Did you compare to other hallucination detection approaches: predictive entropy, semantic entropy, p(true)?

**Ethical Concerns:**

["NO or VERY MINOR ethics concerns only"]

**Final Justification:**

The authors' address my concerns so I maintain my score.

**Limitations:**

Yes

**Quality:**

3

**Strengths And Weaknesses:**

Strengths:

- Novel and well motivated technical contribution.
- Fast and general purpose framework which can be used in practice with minimal overhead after pretraining.
- Strong empirical results showing generalisation and improvement over other probing and logit based methods

Weaknesses:

- No comparison to other hallucination detection approaches: predictive entropy, semantic entropy, p(true)

---

> ### Author Rebuttal · Authors · 2025-07-30
>
> Thank you for your positive feedback. We're pleased that you found our technical contribution novel and well-motivated, and that you appreciated the practicality of our framework and the strength of our empirical results. Below, we address each of the points you raised.
>
> **"No comparison to other hallucination detection approaches: predictive entropy, semantic entropy, p(true)"**
>
> Thanks for pointing this out. In our paper, we compared our approach with methods of low detection latency, and hence mostly focused on approaches based on output scores/probabilities and probing on hidden states.
> The techniques the reviewer is referring to operate on multiple LLM generations or multiple prompts, thus introducing a non-negligible computational overhead that may prevent their real-time use. We note, though, that these may represent strong hallucination detection baselines, and we agree that comparing with them may improve the quality of our contribution. We will further emphasize it in our writing.
> Within the limited time frame of the rebuttal period, we conducted a preliminary experiment on P(true) and Semantic Entropy (SE), in particular. In the table below, we report results for the Movies dataset and two LLMs: LlaMa-8B and Mistral-7B (Mis-7B), which we considered in our paper. Note that SE was evaluated with the Deberta entailment model considered in the original publication [4].
>
> | Method    | Mis-7B – Movies (AUC) |  LlaMa-8B – Movies (AUC) |
> |-------|------|-----|
> | P(True)  | 62.00 | 66.00 |
> | Semantic Entropy | 70.06 | 72.41|
> | **ACT-ViT** | **79.63** | **84.81** |
>
> We note that in all cases, these approaches are significantly outperformed by ACT-ViT, which is not surprising since they are considered “gray-box” (don’t have access to internal artifacts, such as hidden states).
> To provide a clearer, quantitative figure on the complexity incurred by these methods, we calculated the average runtime required by SE to output a prediction. This includes generating $10$ additional responses, and clustering them by calculating mutual entailments via the auxiliary Deberta model. On average, SE required around 6.2 seconds per sample on the Movies/LlaMa combination (std 2.5), and slightly less than 6 seconds on Movies/Mistral (5.9, std 1.7). Note that we made our best efforts to minimise the overhead caused by auxiliary generations by running them in parallel via batching. As a last comment, we note that the time required by the clustering phase is not negligible, as it amounts to ~ 2 seconds on LlaMa responses and ~ 1.35 second on Mistral’s.
> These results concur to underscore the relevance of our approach, which not only performs better, but also much faster, with detection runtimes in the order of $10^{-5}$ seconds, as we noted in our paper (line 75).
>
> **"This is framed as hallucination detection but it seems like the task is accuracy prediction. Could you clarify your definitions?"**
>
> The task we consider is essentially to classify whether the LLM’s response is correct or incorrect. We formulated this task as “hallucination detection”, consistently with previous papers e.g., [1,2,5,6], but we understand the source of confusion as this is very related to accuracy prediction. We will clarify this aspect in the next revision of the paper.
>
> **"Have you investigated other probing tasks?"**
>
> In this paper, we focused specifically on hallucination detection, as we believe it is a sufficiently rich and impactful problem. This focus is also consistent with prior probing works, such as [1, 2], which concentrate only on this task. That said, applying our approach to other probing tasks—such as data contamination detection [3]—is a promising direction, and we consider it an avenue for future work.
>
>  **"Have you looked into interpreting the ACT-ViT itself? What layers/tokens are relevant and whether they agree with the single token/layer results?"**
>
> Thank you for the thoughtful question. While we haven't yet conducted a deep analysis of ACT-ViT’s interpretability, we agree that this is a valuable and promising direction. Understanding why ACT-ViT works could provide important insights into its effectiveness. Since our framework draws inspiration from principles in Vision, we plan to explore interpretability methods such as saliency maps and other techniques like those discussed in [7].
>
>
>  **"Did you compare to other hallucination detection approaches: predictive entropy, semantic entropy, p(true)?"**
>
> We kindly refer the reviewer to the response provided to the first comment.
>
> **References:**
>
> [1] The Internal State of an LLM Knows When It’s Lying. Azaria et al., EMNLP Findings 2023
>
> [2] Llms know more than they show: On the intrinsic representation of llm hallucinations. Orgad et al., ICLR 2024
>
> [3] Detecting Pretraining Data from Large Language Models. Shi et al. ICLR 2024
>
> [4] Detecting hallucinations in large language models using semantic entropy. Farquhar et al., Nature 2024
>
> [5] Language Models (Mostly) Know What They Know. Kadavath et al., Arxiv 2022  (Anthropic AI)
>
> [6] The Curious Case of Hallucinatory (Un)answerability: Finding Truths in the Hidden States of Over-Confident Large Language Models. Slobodkin et al., EMNLP 2023
>
> [7] A Unified Approach to Interpreting Model Predictions. Lundberg et al.,  NeurIPS 2017

---

> > ### Comment · Reviewer_gx9D · 2025-08-03
> >
> > Thank you for your response. My concerns have been addressed. I will maintain my accept score.

---

> > > ### Author Response · Authors · 2025-08-07
> > >
> > > Thank you very much for your positive feedback. We truly appreciate your support and are glad that our responses addressed your concerns.

---

### Official Review · Reviewer_hUsu · 2025-06-30

**Clarity:** 3
**Significance:** 4
**Originality:** 4
**Rating:** 5
**Confidence:** 4

**Summary:**

The paper addresses a research question:
- How to solve the dynamic existence of hallucination information depending on the location of tokens and layers?

Prior work of probing classifiers has shown that the performance of hallucination detection depends on the selected location of tokens and layers. First, the paper shows that each LLM has a different location of tokens and layers for the best performance (See Fig. 2). This finding implies that we need to specific classifier for the given LLM, which is not generalizable to the different LLM.

Based on the findings, the paper defines Activation Tensor (ACT) which has three dimensions: the number of layers, the number of tokens, and the dimension of the hidden state. ACT contains rich information on hallucination within a model and is used for detecting hallucinations. The paper proposes ACT-VIT architecture which takes ACT as input and hallucination classification as output. To leverage data of diverse LLMs, the paper introduces the adapter and pooling to align the spatial and feature dimensions across LLMs.

Experiments are composed of In-domain (ID) and out-of-domain (OOD) setups. In ID, the performance of ACT is larger than the probing methods. Note that probing methods are strong baselines because the classifiers are trained in a supervised way. In OOD, ACT-VIT shows the dataset and LLM generalization; for the LLM generalization, the paper fixes VIT and trains the adaptor. The paper validates the proposed method.

**Questions:**

See above Weaknesses.

**Ethical Concerns:**

["NO or VERY MINOR ethics concerns only"]

**Final Justification:**

The authors addressed my questions, such as the setup details of sequence lengths and dataset size, and the fine-tuning approach of the adaptor. I agree that the reusability of the adaptor is a future research direction, as mentioned in the authors' rebuttal. I have also read other responses. I think that the linear prob is a strong baseline with high performance for HD because it is a training approach. I will maintain my score.

**Limitations:**

Yes, the paper includes Limitations in Sec. 6 and  Broader Impact in Appendix D,

**Paper Formatting Concerns:**

I have no concerns for paper formatting.

**Quality:**

3

**Strengths And Weaknesses:**

### Strengths
- The paper provides the analysis that the location of hallucination information is different depending on LLMs. These findings are connected to the generalization and motivates the design of ACT and ACT-VIT architecture for dynamic existence of hallucination and generalization.
- ACT is large; thus, the paper considers the efficient design to use it.
- Although probing classifiers show promising performance in hallucination, the performance of OOD is low. The paper validates that the proposed method is effective in OOD setup.

### Weaknesses
- Is the sequence length N fixed? Is it zero-padding to match the max sequence length?
- What is the dataset size for pre-training?
- Is the adaptor reusable if the hidden dimensions are aligned?

---

> ### Author Rebuttal · Authors · 2025-07-30
>
> Thank you for your positive feedback. We're pleased to see that you found our analysis valuable—especially our findings on how the location of hallucinated information varies across different LLMs, and the effectiveness of ACT-ViT in the off-domain setting. Below, we will address each of the points you raised.
>
>
> **Is the sequence length N fixed? Is it zero-padding to match the max sequence length?**
>
> Yes, we follow the setup described in [1], where the maximum response length is fixed at 100 tokens. If a response is shorter than this length, we apply zero-padding to reach the maximum length.
>
> **What is the dataset size for pre-training?**
>
> Each LLM-dataset pair consists of 10,000 samples. We have a total of 15 such combinations. Depending on the experimental setup, different subsets of these 15 combinations are used. As a result, the total pre-training dataset size varies between 10,000 and 150,000 samples, depending on the experiments.
>
> **Is the adaptor reusable if the hidden dimensions are aligned?**
>
> Yes, that’s a valid point—thanks for noting it. In principle, given prior knowledge that two LLMs have the same hidden dimension and that their hidden dimensions are aligned (e.g. a finetuned model), they could potentially share the same linear adapter. Experimenting with this aspect could be an interesting future direction.
>
>
>
> **References:**
>
> [1] Llms know more than they show: On the intrinsic representation of llm hallucinations. Orgad et al., ICLR 2024

---

> > ### Comment · Reviewer_hUsu · 2025-08-04
> >
> > I thank the authors for addressing my questions. I have also read other responses, such as an ablation study of ACT-ViT (best token, best layer, last layer). Additional concerns raised by other reviewers are the comparison with other methods. However, I think that the linear prob is a strong baseline with high performance for HD because it is a training approach, so the comparison seems to be sufficient. I will maintain my score.

---

> > > ### Author Response · Authors · 2025-08-07
> > >
> > > We greatly appreciate your positive evaluation and the thoughtful feedback you’ve provided throughout the whole process.

---

### Official Review · Reviewer_Brd8 · 2025-07-02

**Clarity:** 3
**Significance:** 2
**Originality:** 2
**Rating:** 4
**Confidence:** 4

**Summary:**

This paper proposes ACT-ViT, a novel architecture for hallucination detection in Large Language Models (LLMs) that leverages full activation tensors rather than individual token-layer pairs. Traditional probing classifiers are limited by their static, LLM-specific nature and fail to generalize across models. ACT-ViT treats activation tensors as image-like structures and applies a Vision Transformer (ViT)-inspired backbone, enabling joint training across multiple LLMs. The model uses linear adapters to align internal representations from different LLMs and supports efficient zero-shot generalization and fast fine-tuning for unseen models. Extensive experiments across 15 LLM-dataset combinations demonstrate ACT-ViT’s superior performance, scalability, and transferability compared to standard probes and logit-based baselines.

**Questions:**

1.	Please explain the importance and state-of-the-art relevance of probing classifiers in LLM Hallucination Detection (HD). Could you cite some directly related literature? In the related work section, the discussion does not seem to be directly relevant to HD (for example, citation [6]). It appears that probing classifiers are neither a cutting-edge nor a mainstream approach in HD.
2.	The effectiveness of your method may be significantly affected by longer input and output text lengths. Please discuss and experimentally validate the impact of long input and output length on the performance of your approach.
3.	Please provide a deeper analysis and explanation of the phenomena observed in Figure 2. Additionally, how does the proposed method adaptively focus on important activations? It appears that the approach simply relies on brute-force training rather than a truly adaptive mechanism.
4.	For the linear adapter (LA), your method trains a separate adapter for each LLM. If your LA can generalize to different LLMs, then traditional classifiers should also be able to generalize across models. For a new model, how do you select the most suitable pre-trained LA from the existing ones?
5.	In the experimental section, I believe the authors should include the following ablation studies for comparison: (1) Token[*]: Similar to Probe[*], but selects the optimal layer-token combination on the test set. This represents the oracle solution and should yield the best performance, serving as the upper bound for your method, even though it is not feasible in practice. (2) ACT-VIT without complete activations: Select the optimal layer-token combination on the validation set, then replicate this token to form the Activation Tensor (A). The rest of the process (model and training process) remains the same as ACT-VIT. (3) ACT-VIT without complete layers: Select the optimal layer on the validation set, then replicate all tokens from this layer to form the Activation Tensor (A). The rest of the process remains the same as ACT-VIT. (4) ACT-VIT with only the last layer: Select all tokens from the last layer, replicate these tokens L times to form the Activation Tensor (A). The rest of the process remains the same as ACT-VIT. These ablation studies can demonstrate that your motivation (adaptive selection) is responsible for the improvements, rather than additional parameters or training. Additionally, please explain why the Logits and Token methods are missing in the Off-domain Hallucination Detection experiments.

**Ethical Concerns:**

["NO or VERY MINOR ethics concerns only"]

**Final Justification:**

The rebuttal can address most of my concern. The original rating is kept.

**Limitations:**

yes

**Quality:**

3

**Strengths And Weaknesses:**

The strengths of this manuscript are summarized as follows:
1.	The paper addresses an important and timely problem—hallucination detection in LLMs. This is a critical research area with growing relevance, given the widespread deployment of LLMs in high-stakes applications.
2.	The paper provides a clear and accurate analysis of the limitations of existing methods and articulates guiding principles for improvement.
3.	The proposed method is grounded in novel empirical observations revealed through insightful preliminary experiments, which effectively motivate the design of the approach. The method itself is thoughtfully constructed, leveraging structured activation tensors and vision-inspired architectures to address the identified challenges.
4.	The paper is clearly written and well-structured, making it easy to follow.
The weaknesses of this manuscript are summarized as follows:
1.	The paper focuses on a specific sub-direction of hallucination detection—probing classifiers—but does not sufficiently justify the significance of this direction. Although the authors cite several related works, many of them are only loosely connected to probing-based hallucination detection, which weakens the motivation and perceived importance of the chosen research scope.
2.	The proposed method, while effective, appears somewhat straightforward and lacks deeper conceptual innovation.
3.	The experimental section is somewhat limited and lacks strong ablation or comparative studies.

---

> ### Author Rebuttal · Authors · 2025-07-30
>
> We appreciate your positive feedback and are glad you recognize the novelty and relevance of our work. Below, we address your comments.
>
> **“The paper focuses on a specific sub-direction of hallucination detection—probing classifiers—but does not sufficiently justify the significance.[...] explain the importance and state-of-the-art [...]”**
>
>
> We believe probing for HD is important for several reasons. First, growing evidence shows LLMs encode strong signals about output correctness in their hidden states [1,2,3,6,8,9,10]. Second, we note, that several methods build on this by probing them—e.g., [1,2] for generic HD (as we deal with), to steer outputs [5], and to catch flawed reasoning or hallucinations in an unanswerable questions [9,10]. Third, importantly, probing hidden states is much more efficient than inferring the model’s correctness via semantic analyses of its generated text [7]. This makes probing an effective and cheaper alternative to, e.g., Semantic Entropy [11], as evidenced by [7] and our preliminary runtime analyses provided in response to Reviewer **gx9D** below. We thank you for this question — we'll revise the Related Work section to highlight this better and add any missing citations.
>
>
> **“The proposed method, while effective, appears somewhat straightforward [...]”**
>
> The main contribution of this work consists of three parts. (1) We identify a specific limitation of existing probing techniques (Section 4, lines 134–167), namely their limited effectiveness in cross-LLM and cross-dataset applications. (2) We propose a solution to this limitation by introducing activation tensors as a new data type, and develop ACT-ViT—an architecture designed to leverage the structural properties of activation tensors to effectively process them. (3) We demonstrate that ACT-ViT not only offers a stronger solution to probing in standard settings (see Table 1 as an example), but also enables novel cross-LLM and cross-dataset applications for HD—including zero-shot generalization to unseen datasets—an area that, to the best of our knowledge, has received little attention (see Figure 4).
>
> **“The effectiveness of your method may be significantly affected by longer input and output text lengths [...]”**
>
> We appreciate the opportunity to clarify. (1) our method detects hallucinations using only the LLM's response, not its input; and (2) ACT-ViT efficiently handles variable-length outputs via dynamic pooling, producing a fixed-size activation tensor. This makes it feasible to process even longer outputs while controlling computational cost.
> To further justify this, and validate ACT-ViT’s robustness to longer outputs, we took your advice and conducted an additional experiment on the Math [4] dataset, which involves processing longer output text sequences [2].  We benchmarked ACT-ViT against all baselines, reporting only the best variant from each class due to space limits – best-probas for probability/logit-based methods and Probe[*] for probing-based ones – results are shown below. ACT-ViT remains the top performer, surpassing all baselines—by ~7 AUC points for LLaMA-8B. We will add this experiment to our paper, along with a proper discussion on the impact of the output length on ACT-ViT.
> |**Method** |**Qwen-7B – Math**| **Mis-7B – Math** |**LlaMa-8B – Math**|
> |-|-|-|-|
> |Best-Probas| 79.12| 56.00| 75.00|
> |Probe[*]| 92.13| 75.60 | 78.81|
> |**ACT-ViT(s) (Ours)**|**92.51** | **76.50**| **85.70** |
>
> **“Please provide a deeper analysis and explanation of the phenomena observed in Figure 2. [...] how does the proposed method adaptively focus on important activations? [...]”**
>
> Thanks for the opportunity to clarify. We note that for a given LLM, the optimal probing token/layer position often varies across datasets [1,2]. On top of this, Figure 2 in our paper shows that even with a fixed dataset, the most predictive layer-token position for HD can differ across LLMs. As discussed in Section 4 (lines 134–174), this highlights a key limitation of standard probing methods in cross-LLM applications, as considered in our work.
> Our method is designed to overcome this limitation by side-stepping the reliance on a single, fixed probing position. By drawing an analogy with images and vision architectures (see Section 4.1), we propose to process the entire Activation Tensor. This allows the model to learn to combine predictive cues from multiple positions, like how vision models detect features across an image to identify faces. In this sense, our approach is “adaptive.” The ablation studies discussed below further validate this design and illustrate its effectiveness; see the relevant section below for details.
>
> **”For the linear adapter (LA), your method trains a separate adapter for each LLM [...]”**
>
> Thanks for raising this point. As the reviewer notes, each LLM has its own Linear Adapter (LA). ACT-ViT is able to effectively learn on Activation Tensors from multiple LLMs as these LAs project their respective hidden states into a common embedding space. Here, the shared ViT Backbone (ViT-B) operates (see Figure 1) – HD is performed in this common space. This facilitates cross-model generalization: for a new LLM only its specific LA is trained, while the ViT-B remains frozen. Results in Table 2 demonstrate that this approach is indeed more effective than probing.
>
> **”[...] I believe the authors should include the following ablation studies for comparison:[...]”**
>
> Thank you for the suggestion. We conducted the requested experiments and present the results below. Before that, we would like to note that selecting a single, optimal token-layer combination for probing across multiple LLMs is ill-defined, as they are not necessarily in correspondence.
>
> That said, your proposed ablations offer valuable insights in a more controlled setting using a single LLM across various tasks, hence, we trained jointly on activation tensors from one LLM across all five datasets—HQA, Movies, HAQ-with-context, IMDB, and TQA—and report results below for Qwen-7B and LlaMa-8B.
>
> In the table:
> - (1) Oracle Probe – probing applied to the best layer-token pair, when selected based on the test set;
> - (2) ACT-ViT (best token) — we select the best token across layers and replicate (broadcast) this 1D signal to form the whole activation tensor;
> - (3) ACT-ViT (best layer)— as above, but we select the best layer across all tokens.
> - (4) ACT-ViT (last layer) — as above but we select the last layer across all tokens.
>
>
>
> |**Method** |**Qwen -7B - HQA**|**Qwen -7B - Movies**|**Qwen -7B - HQA-WC** |**Qwen -7B - IMDB**|**Qwen -7B - TQA**|**Average**|
> |-|-|-|-|-|-|-|
> | Probe[*]   | 80.57  | 93.22  | 75.01 | 93.76  | 87.08 |85.93|
> | (1) Oracle Probe | 81.97  | 93.36 | 75.26  | 93.97  | 87.41 | 86.39 |
> | (2) ACT-VIT (best token) | 59.97  | 72.92  | 55.46  | 53.70  | 62.86 | 60.98|
> | (3) ACT-VIT (best layer)  | 84.86   | 92.38  | 75.06| 93.18 | 88.20 |86.74 |
> | (4) ACT-VIT (only last layer)| 82.36 | 92.90 | 71.70 | 92.52 | 86.32 | 85.16 |
> | **ACT-ViT**  | **88.20**  | **95.79** | **78.19** | **96.22** | **91.34** |**89.95** |
>
>
> | **Method**| **LlaMa - 8B - HQA** | **LlaMa - 8B - Movies** | **LlaMa - 8B - HQA-WC** | **LlaMa - 8B - IMDB** | **LlaMa - 8B - TQA** | **Average** |
> |-|-|-|-|-|-|-|
> | Probe[*] |79.71 | 83.61|70.56| 93.58|81.09 |81.71|
> | (1) Oracle Probe | 79.84 | 83.61 | 72.80| 93.68| 81.72|82.33  |
> | (2) ACT-VIT (best token) | 77.39| 74.01| 59.79 | 78.02 | 77.33 |73.31  |
> | (3) ACT-VIT( best layer)| 82.16  | **83.91** | 71.94 | 92.70| 83.83 | 82.91 |
> | (4) ACT-VIT (only last layer) | 80.51 | 80.57| 69.64 | 91.90 | 81.15 | 80.75  |
> | **ACT-ViT**| **82.22** | 83.70  | **72.71**|**93.55** | **84.68** |**83.37** |
>
>
> **Observations.**
>
> - (O1) Omitting the Oracle Probe, ACT-ViT is the best performing method overall – in 9/10 cases (see Bold in the Table). This suggests that processing the whole Activation Tensor is an advantageous strategy.
> - (O2) ACT-ViT(best layer) is second best, better than ACT-ViT (best token), indicating that committing to a single layer is more effective than focusing on a single token.
> - (O3)  As expected, the Oracle Probe outperforms the standard Probe. However, in most cases, it is notably outperformed by ACT-ViT. Note that the Oracle Probe is not an upper bound to ACT-ViT: ACT-ViT can, in fact, aggregate and combine information from different positions together (unlike standard probing).
>
>
> **”[...] please explain why the Logits and Token methods are missing in the Off-domain [...]”**
>
>
> To clarify, both methods are used off-domain, but only the best variant of each is shown: best-probas (logit-based) and Probe[*] (probing). Full results will be added to the appendix.
>
> **References:**
>
> [1] The Internal State of an LLM Knows When It’s Lying. Azaria et al., EMNLP Findings 2023
>
> [2] Llms know more than they show: On the intrinsic representation of llm hallucinations. Orgad et al., ICLR 2024
>
> [3] Language Models (Mostly) Know What They Know. Kadavath et al., Arxiv 2022  (Anthropic AI)
>
> [4] Benchmarking hallucination in large language models based on unanswerable math word problem. Sun et al., CoRR 2024
>
> [5] Inference-Time Intervention: Eliciting Truthful Answers from a Language Model. Li et al., NeurIPS 2023
>
> [6] InternalInspector $I^2$: Robust Confidence Estimation in LLMs through Internal States. Beigi et al., EMNLP 2024
>
> [7] Efficient and Effective Uncertainty Quantification for LLMs. Xiong et al., SafeGenAI NeurIPS Workshop 2024
>
> [8] Discovering Latent Knowledge in Language Models Without Supervision. Burns et al., ICLR 2023
>
> [9] Reasoning Models Know When They're Right: Probing Hidden States for Self-Verification. Zhang et al., Arxiv 2025
>
> [10] The Curious Case of Hallucinatory (Un)answerability: Finding Truths in the Hidden States of Over-Confident Large Language Models. Slobodkin et al., EMNLP 2023
>
> [11] Detecting hallucinations in large language models using semantic entropy. Farquhar et al., Nature 2024

---

### Note · Authors · 2025-08-11

We thank all reviewers for their positive and constructive feedback. Before scores were hidden, ratings were **gx9D - 5**, **hUsu - 5**, **Brd8 - 4**, and **kwZz - 3**, averaging 4.25.

The reviewers acknowledged the novelty of our approach, its relevance for future research, and the significance of the task we address, i.e., hallucination detection in LLMs:

- “The paper addresses an important and timely problem—hallucination detection in LLMs” (Brd8 - 4)
- “Novel and well motivated technical contribution” (gx9D - 5)
- “The paper presents [..] which may be useful for future research” (kwZz - 3)
- “The proposed method is grounded in novel empirical observations [..]” (Brd8 - 4)

Our approach considers activations across layers and generation steps in a structured manner and is natively designed to operate across LLMs and datasets . The reviewers appreciated these aspects, as well as our motivating experimental analyses.

- “The paper provides the analysis that the location of hallucination information is different depending on LLMs. [..] motivates the design of ACT and ACT-VIT [..]” (hUsu - 5)
- “Strong empirical results showing generalisation and improvement over other probing and logit based methods” (gx9D - 5)
- “Although probing classifiers show promising performance in hallucination, the performance of OOD is low. The paper validates that the proposed method is effective in OOD setup” (hUsu - 5)
- “Across 15 LLM-dataset pairs, ACT-ViT consistently outperforms prior white-box models and supports zero-shot generalization to new datasets” (kwZz - 3)

We fully addressed all constructive feedback in our rebuttal, as acknowledged by the reviewers:

- Brd8 - 4: “Thanks for the response, which solved my concerns. I decide to keep my positive score”
- hUsu - 5: “I thank the authors for addressing my questions. I have also read other responses [..] I will maintain my score”
- gx9D - 5: “[..] My concerns have been addressed. I will maintain my accept score”

Reviewer kwZz - 3 questioned the necessity of hallucination detection, stating that their *“biggest concern lies in the necessity of this task, rather than the authors’ implementations.”* We provided detailed justification for the value of this task, after which the reviewer indicated they would *“calibrate with other reviewers who have more knowledge”* on the topic. We hope such calibration occurs, as the importance of hallucination detection has been reinforced by the other reviewers (see bullets above).

---

### Decision · Program_Chairs · 2025-09-17

**Decision:**

Accept (poster)

**Comment:**

(a) Summary of Claims and Findings

This paper introduces ACT-ViT, a novel method for hallucination detection that addresses the poor generalization of prior probing classifiers. By treating an LLM's full activation tensor as an image-like structure processed by a ViT-based architecture, the model can be trained jointly across different LLMs. The key finding is that this approach significantly outperforms baselines and demonstrates strong zero-shot generalization to unseen models and datasets.

(b) Strengths

The paper's core strengths are its novel and well-motivated approach to the critical problem of LLM hallucination, backed by strong empirical results that validate its superior performance and generalization capabilities. The work is technically sound and clearly presented.

(c) Weaknesses

Initial weaknesses, primarily a limited comparison to non-probing baselines (e.g., Semantic Entropy) and the need for more detailed ablations, were comprehensively addressed during the author rebuttal period.

(d) Reasons for Acceptance

The primary reason for acceptance is the paper's elegant and impactful solution to a key challenge in LLM reliability. The ACT-ViT method is a technically sound and novel contribution that demonstrates impressive generalization—a significant advance for probing-based techniques. The thorough empirical validation solidifies this as a strong and valuable contribution to the field.

(e) Summary of Discussion and Rebuttal

The rebuttal period was highly productive. The authors provided extensive new experiments, including comparisons against requested baselines and a full suite of ablation studies. This successfully addressed all major technical concerns, converting reviewer feedback into a strong positive consensus. The paper was substantially improved by the review process and is now in excellent shape for publication.